# Screening of Yeasts Isolated from *Baijiu* Environments for Producing 3-Methylthio-1-propanol and Optimizing Production Conditions

**DOI:** 10.3390/foods11223616

**Published:** 2022-11-12

**Authors:** Jinghao Ma, Liujie Cheng, Yujiao Zhang, Yuchun Liu, Qi Sun, Jie Zhang, Xiaoyan Liu, Guangsen Fan

**Affiliations:** 1School of Food and Health, Beijing Technology and Business University (BTBU), Beijing 100048, China; 2Academy of National Food and Strategic Reserves Administration, Beijing 100037, China; 3Beijing Engineering and Technology Research Center of Food Additives, Beijing Technology and Business University (BTBU), Beijing 100048, China

**Keywords:** *Baijiu*, 3-Methylthio-1-propanol, L-methionine, optimization, *Hyphopichia burtonii*, aroma production

## Abstract

3-Methylthio-1-propanol (3-Met) is widely used as a flavoring substance and an essential aroma ingredient in many foods. Producing 3-Met by microbial transformation is green and eco-friendly. In the present study, one strain, YHM-G, which produced a high level of 3-Met, was isolated from the *Baijiu*-producing environment. Strain YHM-G was identified as *Hyphopichia burtonii* according to its morphological properties, physiological and biochemical characteristics, and ribosomal large subunit 26S rRNA gene D1/D2 domain sequence analysis. The optimal conditions for 3-Met production by YHM-G were obtained by single factor design, Plackett–Burman design, steepest ascent path design and response surface methodology as follows: 42.7 g/L glucose, pH 6, 0.9 g/L yeast extract, 6 g/L L-methionine (L-Met), culture temperature 28 °C, shaking speed 210 rpm, loading volume 50 mL/250 mL, inoculum size 0.5% (*v/v*), culturing period 48 h and 2.5 g/L Tween-80. Under these optimal conditions, the 3-Met production by strain YHM-G was 3.16 g/L, a value 88.1% higher than that before optimization. Strain YHM-G can also produce a variety of flavor compounds that are important for many foods. This strain thus has the potential to increase the abundance of 3-Met in some fermented foods and enhance their aroma profiles.

## 1. Introduction

3-Methylthio-1-propanol (3-Met), a sulfur-containing higher alcohol, is a crucial aroma component of many foods and beverages, such as soy sauce, cheese, *Baijiu*, beer and wine and is naturally present in apples, tomatoes and other foods [1]. 3-Met has a low odor threshold of 1–3 ppm, with a recommended dosage in food of 0.1–10 mg/kg [2]. It has been classified by the US Federal Emergency Management Agency in the class of safe and edible flavor substances. It is widely used in flavor and fragrance formulation and food flavoring because it imparts cauliflower-like, meaty, savory or toasted cheese flavor notes [3].

At present, 3-Met is produced by chemical synthesis and biotechnological methods [3]. Chemical synthesis is mainly used because of its lower costs but it produces undesired side products, resulting in the formation of racemic mixtures and environmental pollution [3,4]. With the increasing demand of consumers for natural products, developing healthier and more eco-friendly natural products have become more important [1,3,4]. Microbial fermentation is a necessary process that uses microorganisms to transform raw materials into products needed by human beings through specific metabolic pathways under appropriate conditions [5]. Recent studies have shown that microorganisms, especially yeasts, can biotransform L-methionine (L-Met) to produce 3-Met by fermentation, an effective method of natural production [3,5]. L-Met is transaminated and then decarboxylated to the corresponding aldehydes via the Ehrlich pathway, then reduced to higher alcohols or oxidized to fusel acids [1]. In recent years, biological conversion methods have become an important development direction for producing natural 3-Met, because these methods could avoid undesired by-products and are considered as more eco-friendly [3,5]. Thus, interest in the bioproduction of 3-Met by fermentation has been growing in recent years.

However, relatively few studies have reported on the ability of microorganisms to produce natural 3-Met, which mainly included a few types of yeast, such as *Saccharomyces cerevisiae*, *Kluyveromyces lactis*, *Debaryomyces hansenii*, *Yarrowia lipolytica*, *Zygosaccharomyces rouxi* and *Williopsis* species [6,7,8,9]. The ability of these natural strains to produce 3-Met is limited with yields generally less than 0.5 g/L (Table 1), therefore their application as a microbial factory in industrial fermentation has been restricted [6,8,10].

With the demand for natural 3-Met increasing and the rapid development of genetic engineering technology, most studies have begun to use molecular biology techniques to modify genes to engineer strains with a high yield [1,16,17]. After determining the key enzymes that affect the synthesis of 3-Met in yeast strain, the yeast genome was modified through gene overexpression or knockout [1,16,18]. The yield of 3-Met from certain genetically modified strains has increased compared with the yield of unmodified strains, with yields of up to 4.38 g/L after fermentation condition optimizations [18,19]. As some consumers may consider 3-Met produced by genetically modified strains unnatural, natural strains with a high 3-Met yield need to be identified and their fermentation conditions should also be optimized to obtain acceptably natural 3-Met production capability.

Therefore, the objective of this study was to obtain a strain with a high 3-Met yield from the *Baijiu*-making environment and to optimize its fermentation conditions. In this study, a yeast identified as *Hyphopichia burtonii* with a high yield of 3-Met was obtained from the *Baijiu*-brewing environment. Several statistical methods will be combined to optimize the fermentation conditions for this strain to further enhance its 3-Met production. To the best of the author’s knowledge, the strain discovered in this study is among the highest 3-Met yielding natural strains previously reported.

## 2. Materials and Methods

### 2.1. Materials, Reagents and Media Preparation

*Daqu* (a starter for *Baijiu* production) samples were collected from different *Baijiu*-making enterprises to screen yeast strains. 3-Met, L-Met and chromatographic grade methanol were purchased from Sigma-Aldrich (St. Louis, MO, USA). All other chemicals were analytical grade and commercially available unless otherwise stated. Yeast extract peptone dextrose (YPD) medium for screening and culturing strains, Wallerstein laboratory nutrient agar (WL) medium for strain morphology identification and Sorghum hydrolysate medium (SHM) for aroma compounds production were prepared as described previously [20]. The initial fermentation medium for 3-Met production contained per liter: 30 g glucose, 0.8 g yeast extract powder, 4 g L-Met, 8 g KH_2_PO_4_, 6 g K_2_HPO_4_, 2 g NaCl, 0.03 g ZnSO_4_, 0.01 g MgCl_2_ and 0.02 g FeCl_2_, pH 5.0. All media were autoclaved at 115 °C for 20 min.

### 2.2. Screening for Yeast with a High Yield of 3-Met

Yeasts were isolated as reported previously [20]. One gram of crushed *Daqu* sample was added to 9 mL sterilized ddH_2_O, then the mixture was diluted to 10^−5^, 10^−6^ and 10^−7^. After dilution, 0.1 mL suspensions with different concentrations were spread onto YPD plates and inverted cultured at 30 °C for 5 days. Single colonies were picked and streaked on YPD plates to obtain a pure culture. After being identified as a pure culture by microscopic examination, each isolated colony was stored at −80 °C. Each yeast strain was pre-cultured in YPD liquid medium, then 0.1 mL activated yeast cell solution, which was adjusted to a concentration of 1 × 10^6^ cells/mL by using a blood cell counting plate, was inoculated into 50 mL fermentation medium and cultured in a 28 °C incubator at 180 rpm for 48 h to evaluate its ability to produce 3-Met. After fermentation, the cultures were centrifuged at 4 °C for 10 min at 13,751× *g*, the clear supernatants were filtered through a 0.22-µm filter and 3-Met production was analyzed by high performance liquid chromatography (HPLC). Each yeast strain was analyzed in parallel three time. The yeast that produced the largest amount of 3-Met was selected for further characterization.

### 2.3. Identification and Biochemical Characteristics of YHM-G

The target yeast strain YHM-G was identified by morphological and physiological characteristics and ribosomal large subunit 26S rRNA gene D1/D2 domain sequence, as described by Fan et al. [20]. YHM-G was examined for colony morphology on the WL plate by the naked eye and cell morphology using a microscope (Nikon, Tokyo, Japan), respectively. Carbon and nitrogen source assimilation and sugar fermentation tests were conducted in stationary cultures as a sole carbon or nitrogen source. Four slanted milliliters of urea medium, which were placed in slants, were applied to the incubated yeast colony at room temperature. Red-pink color in the medium was considered to be a positive test for urea test. For the indole test, the appearance of bright red and yellow color composed after adding 0.5 mL of Kovac’s reagent to incubated culture at 30 °C for 24 h on indole test media indicated positive and negative results, respectively. After adding methyl red indicator solution to inoculated culturing media and incubating at 30 °C for up to 7 days, the changing color to red indicates that the methyl red test is positive. For the Voges–Proskauer test after incubation of 48 h Voges–Proskauer Reagent A (5% naphthol) and Voges–Proskauer Reagent B (40% KOH) were added and a change in color was observed. YHM-G was inoculated on triple sugar iron agar slant and cultured at 30 °C for 48 h, and results were determined based on color change. Citrate test was performed via inculcating Simmons Citrate Agar plates surface with yeast cultures, then incubating at 30 °C up to 48 h, and the media color changing from green to bright blue indicates a positive reaction. A gelatin liquidized test was applied; when YHM-G was inoculated into tubes containing gelatin, gelatin liquefaction was the positive result. To confirm the starch hydrolysis activity, YHM-G was grown on nutrient agar containing 1% soluble starch and incubated at 30 °C for 72 h. Hydrolysis was observed as the formation of a colorless zone around the colony after flooding the plate with Gram’s iodine. The ribosomal large subunit 26S rRNA gene D1/D2 domain fragment from strain YHM-G was amplified using universal primers (NL1: 5′-GCATATCAATAAGCGGAGGAAAAG-3′ and NL4: 5′-GGTCCGTGTTTCAAGACGG-3′), as described by Kurtzman and Robnett [21]. Then, the resulting sequence was submitted to the NCBI database (https://www.ncbi.nlm.nih.gov/, accessed on 17 July 2022) with accession number ON999037 and a BLAST search was conducted against the Nucleotide collection (nr/nt) database using the BLASTN program. All reference sequences were from type or otherwise authenticated strains. Moreover, the overlapping fragment of D1/D2 domain of the ribosomal large subunit 26S rRNA gene was then aligned using ClustalW. The sequence alignment was filtered in ARB to exclude gaps, missing data, ambiguous nucleotide positions and lowercase positions. A phylogenetic tree was constructed by the neighbor-joining method (MEGA version 7 software) using the Kimura 2-parameter model [22] and tested by bootstrap with 1000 repetitions [23].

The growth response was measured in a certain range of parameters, including temperature (20, 25, 30, 35, 40, 45 and 50 °C), pH (1–14), glucose concentration (23, 29, 33, 38, 41, 44 and 47%, *w/w*), NaCl concentration (0, 3, 6, 8, 11, 13, 15, 17, 19 and 21%, *w/v*), ethanol concentration (0, 3, 6, 9 12, 15, 18 and 21%, *v/v*) and 3-Met concentration (5, 10, 20, 30, 35, 40, 45 and 50 g/L) by monitoring the optical density at 560 nm (OD_560_) of the YPD medium, as reported previously [20].

The aroma compounds produced by YHM-G in SHM were carried out as described previously [24].

### 2.4. Optimization of Culture Conditions by Single Factor Design

After preculture in 30 mL YPD at 28 °C for 24 h, 0.1 mL of YHM-G cell suspension was inoculated into the fermentation medium as described above. Various conditions, including glucose concentration, yeast extract concentration, L-Met concentration, the time point of L-Met addition, initial pH, temperature, shaking speed, loading volume, inoculum size, surfactant types, Tween-80 concentration and time, were optimized for 3-Met production under submerged fermentation using single factor design (Appendix A).

### 2.5. Optimization of Culture Conditions by Plackett-Burman (PB) Design

According to the results from the single factor design, eleven variables, including shaking speed (X_1_), temperature (X_2_), glucose concentration (X_3_), initial pH (X_4_), loading volume (X_5_), inoculum size (X_6_), the time point of L-Met addition (X_7_), culture time (X_8_), yeast extract concentration (X_9_), L-Met concentration (X_10_) and Tween-80 concentration (X_11_) were chosen for PB design to screen important variables for 3-Met production (Table 2). Each variable was represented at two levels denoted by (+1) and (−1), respectively. The order of the experiments was fully randomized.

### 2.6. Optimization of Culture Conditions by Steepest Ascent Path Design

The steepest ascent path design was constructed with important variables screened by the regression model in PB design. The direction of the steepest ascent path for each variable was determined according to the regression results in PB design and the steep path for each variable was determined by the regression model from the PB design and practical experience (Table 3). Six experiments were conducted in the steepest ascent path design.

### 2.7. Optimization of Culture Conditions by Response Surface Methodology (RSM) Design

After the center point identified by the steepest ascent path design, RSM was used to optimize three variables screened according to statistically significant influence on 3-Met production by PB design based on the Box–Behnken experimental design (BBD, Design-Expert Software 11.0, StatEase Inc., Minneapolis, MN, USA). The design with three factors (time point of L-Met addition (A), glucose concentration (B) and yeast extract concentration (C)) and three levels was listed in Table 4, including three replicates at the center point.

### 2.8. Analytical Methods

3-Met concentration were determined using a 1260 series HPLC (Agilent, Santa Clara, CA, USA) equipped with a ZORBAX Eclipse Plus C18 column (4.6 × 250 mm, 5 µm) and a UV detector [16]. After the injection of 10 µL culture sample, the column was eluted with methanol-water (30: 70, *v/v*) at 30 °C at a flow rate of 0.7 mL/min for 15 min. The UV-detector was used at a fixed wavelength of 215 nm. The chromatogram grade 3-Met (Sigma-Aldrich Chemical Co., St. Louis, MO, USA) was prepared by adding 0, 1, 2, 3, 4 and 5 mg of the standard solutions (0–5 g/L) in 1.0 mL of ultrapure water into a serial concentration to set up the calibration curve. The standard calibration curve working solution was analyzed by HPLC. The calibration curve was obtained and used to calculate of 3-Met in the sample. The analysis was performed in triplicate. The aroma compounds were analyzed by headspace solid–phase microextraction gas chromatography–mass spectrometry (HS–SPME–GC–MS) with a TraceGC TSQ 8000 Evo instrument (Thermo Fisher Scientific, Waltham, MA, USA) according to our previous report [25]. Briefly, the aroma compounds in samples were extracted by 50/30-µm divinylbenzene/carboxen on polydimethylsiloxane (DVB/CAR on PDMS)-coated fibers. After centrifuge, 5 mL supernatant, 2 g NaCl and 10 µL of 0.5 g/L tetraoctanol solution were placed into a 15-mL headspace vial. After being kept in a water bath at 50 °C for 10 min, a SPME needle was inserted into the vial to extract aroma compounds for 30 min at 50 °C. Then, the extraction head was inserted into the GC-MS inlet for desorption at 250 °C for 5 min. The GC-MS conditions were as follows: 40 °C for 3 min, increased up to 100 °C for 5 min, then 150 °C for 3 min and to 280 °C for 6 min. The impact energy was 70 eV, and the mass spectral scanning range was *m/z* 33–550. The splitless mode was conducted for these analyses. The aroma compounds that the mass spectra with positive and negative matches higher than 800 with the National Institute of Standards and Technology library are reported. The amount of each aroma compound was calculated as the ratio of the mass concentration of tetraoctanol to the mass concentration of the compound.

### 2.9. Statistical Analysis

Statistical differences in the assessed strategies were analyzed using a one-way ANOVA (*p* < 0.05) with Tukey’s test. Assays have been conducted at least in triplicate, and the reported values correspond to the mean value and standard deviation. SPSS 24.0 (IBM Corp., New York, NY, USA), OriginPro 9.1 (OriginLab, Northampton, MA, USA), Design-expert 11 (Stat-Ease, Inc., Minneapolis, MN, USA) and Excel 2019 (Microsoft, Redmond, WA, USA) were used to process and analyze data.

## 3. Results and Discussion

### 3.1. Yeast Screening for 3-Met Production

A total of 46 yeasts were isolated from *Daqu* samples then their ability to produce 3-Met was tested. Several of these yeasts were able to produce 3-Met but the yield of most strains was relatively low, as confirmed in other reports (Table 1), so they had no potential to produce natural 3-Met [2,6,16,26]. In the present study, four yeasts produced 3-Met at concentrations exceeding 1.0 g/L (Table 5), which gave them an advantage over most previous yeasts, such as *S. cerevisiae* EC-1118, *K. lactis* KL71, *R. toruloides* 6740 and *R. acuta* 7028 (Table 1) [2,3,13]. Among the high 3-Met yielding strains evaluated in this study, the 3-Met yield of YHM-G was not significantly higher than strain F3301 but was significantly higher than strain Y1801 and F9403, and the yeast strain YHM-G produced 1.68 g/L of 3-Met among the highest 3-Met yield in natural microorganisms (Table 1) [27]. Strain YHM-G was therefore chosen as the target strain for the optimization of the medium composition and cultivation conditions to improve the low 3-Met productivity in natural strains.

### 3.2. Identification of Strain YHM-G

After growing for 48 h at 30 °C on WL plates, the colonies of strain YHM-G were white, raised, with dry and rough surfaces and smooth edges and not easy to pick up (Appendix A). The WL medium had changed from blue to yellow where the yeast had grown (Appendix A). The cells of YHM-G observed under a microscope were rodlike with asexual budding reproduction occurring at one end of the cells without an endospore and mycelium (Appendix A). Thus, strain YHM-G exhibited the morphological and cellular structure of yeast.

The physiological characteristics of strain YHM-G are shown (Appendix A). The sugar fermentations showed that strain YHM-G could use most types of sugar. Acid production and aerogenesis were observed when strain YHM-G was fermented with glucose, maltose, D-galactose and sucrose. Strain YHM-G only produced acid from D-maltose and aerogenesis from D-xylose and was unable to ferment L-rhamnose monohydrate, D-arabinose and lactose, as indicated by the lack of acid or gas production. Strain YHM-G was able to utilize glucose, maltose, D-galactose, sucrose, D-maltose, D-xylose, ethanol, glycerol, inulin, L-rhamnose monohydrate, D-arabinose and lactose but not D-raffinose, D-trehalose, mannose, D-ribose and D-sorbose. All nitrogen sources tested, including urea, ammonium sulfate, sodium nitrate, potassium nitrite, L-phenylalanine and L-lysine, were able to be used as the sole nitrogen source for growth.

The YHM-G strain was positive for the indole test, methyl red test, starch hydrolysis test, citrate test and urea test, indicating that it was able to produce indole from tryptophan, acid from glucose and amylase to decompose starch and make use of sodium citrate or urea as a carbon source or nitrogen source for growth (Appendix A). Additionally, the Voges–Proskauer test, hydrogen sulfide test and gelatin liquification test were negative. From these physiological and biochemical characteristics, the YHM-G strain was preliminarily confirmed as *Hyphopichia* sp. These results were helpful in understanding its metabolic activities, so that its metabolic pathway could play its corresponding functions.

After searching the NCBI database, the ribosomal large subunit 26S rRNA gene D1/D2 domain fragment amplificated from YHM-G showed a 100% sequence similarity with a partial 26S rRNA gene sequence from *H. burtonii* (CP024760.1, MH867400.1, KY107885.1, KY107883.1, KY107882.1, KY107877.1, HF952839.2, KC192660.1 and MK373312.1). According to branch clustering, strain YHM-G was closest to *H. burtonii* CBS:2352 (KY107882.1) and *H. burtonii* CBS:232.37 (MH867400.1) in the phylogenetic tree constructed using MEGA 7.0.14 software (https://www.megasoftware.net, accessed on 17 July 2022) based on the neighbor-joining statistical method (Appendix A). Overall, strain YHM-G was identified as *H. burtonii*, according to the analysis of morphological, physiological and biochemical characteristics, and phylogenetic analysis. Although there is relatively little research on this strain, it has been reported to exist in the production process of many fermented foods, such as *Baijiu* [28], *Huangjiu* [28], ham [29], *Douchi* [30], Douban sauce [31], Kombucha [32], wine [33], traditional sourdough starters [34] and cocoa beans [35] through the use of traditional screening and high-throughput sequencing methods. Its presence has most often been reported in *Baijiu*, and there have been few studies on the role of this strain in other fermented foods [28,36]. To enhance research on the role of yeasts other than *S. cerevisiae* in food, the functions of strain YHM-G will be continuously analyzed.

### 3.3. Tolerance Features of YHM-G

3-Met not only has outstanding flavor characteristics, which endow specific flavor characteristics to traditionally brewed food products, such as *Baijiu*, but also offers the healthy function of antioxidant activity [7,37]. Therefore, increasing the content of 3-Met in traditionally brewed foods, such as *Baijiu*, will help to improve their quality. Strain YHM-G has been identified as a yeast with a high yield of 3-Met preliminarily screened from the *Baijiu* production environment. Whether it can be applied to the *Baijiu* production to improve its 3-Met content depends on its biological characteristics because *Baijiu* production takes place in a constantly changing microenvironment. Only the strain YHM-G was found to offer excellent environmental tolerance and survivability for a long time in the *Baijiu* production process, allowing it to produce 3-Met and improve *Baijiu* quality. Therefore, we analyzed the adaptability of strain YHM-G to the environment regarding its range of growth temperatures and pH and tolerance to sugar, NaCl, ethanol and 3-Met (Figure 1). The optimum growth temperature of strain YHM-G was 25 °C, similar to most yeasts found in previous reports, with a maximum growth temperature of 40 °C (Figure 1a), which was higher than most temperatures (20–40 °C) during the *Baijiu* production process [38,39,40,41,42]. Therefore, strain YHM-G can adapt well to the different temperatures used in *Baijiu* production. Strain YHM-G was able to grow between pH 2 and pH 12 (Figure 1a), a range wider than previously reported [42]. Its optimum pH for growth was weak acidic, at pH 4–7 (Figure 1a). This range is within the range of pH used in the *Baijiu*-production process [38,39,40]. As the glucose concentration, sucrose concentration and NaCl concentration increased the growth rate of strain YHM-G declined. When the concentration of glucose, sucrose and NaCl exceeded 47%, 47% and 17%, respectively, strain YHM-G could not grow (Figure 1b). In general, yeast strains have a high tolerance to sugar with strains, such as *Pichia kudriavzevii* YF1702 and *Clavispora lusitaniae* YX3307 [25,43,44,45,46]. Additionally, like other microorganisms, yeasts have a limited tolerance to NaCl, a concentration of 15% preventing the growth of most microorganisms [42,47]. Compared with most other reports on yeast, strain YHM-G had a high tolerance to NaCl, so it could better adapt to different environments [47,48]. However, there are a few yeasts with a high tolerance of up to 30% NaCl screened from particular environments, possibly related to the activation of antioxidant enzymes in the antagonistic yeast, including catalase, thioredoxin reductase, glutathione peroxidase and glutathione reductase [49,50]. As the ethanol concentration increased, the biomass concentration of strain YHM-G increased first and then decreased. At a concentration of ≤6%, ethanol could be used as a carbon source for the growth of strain YHM-G, consistent with the previous physiological and biochemical results (Figure 1b and Appendix A). At an ethanol concentration of ≥6%, the growth of strain YHM-G was inhibited because of the influence of ethanol on the cell structure and enzyme activity of the strain (Figure 1b) [25,51]. The ethanol concentration was 12% at maximum growth (Figure 1b), a value much higher than that produced during *Baijiu* production processes (2–4%), thus strain YHM-G can be considered a yeast with high ethanol tolerance [51,52]. The ability of a strain to tolerate 3-Met determines its level of 3-Met production. The results showed that strain YHM-G could grow at a 3-Met level of 40 g/L (Figure 1c), similar to that of *S. cerevisiae* CEN.PK113-7D, indicating that strain YHM-G has the potential to produce more 3-Met [53]. Overall, strain YHM-G can adapt well to different environments for the traditional brewing foods, such as *Baijiu*. Therefore, strain YHM-G could be used for brewing these foods and increasing their contents of 3-Met.

### 3.4. Optimization of Culture Conditions by Single Factor Design

Although producing 3-Met by yeasts from sugars via de novo biosynthesis has been rare, previous reports have emphasized that its biosynthesis by yeasts requires sugars [2,3]. Therefore, the effect of glucose concentration on 3-Met production was evaluated: it increased initially as the glucose concentration increased and then decreased at higher concentrations. The yield of 3-Met was highest when the concentration range was 30–60 g/L (Appendix A), which agreed with previous reports [19]. When glucose is used as a carbon source, it provides energy for yeast growth and its concentration influences the rate of growth and the metabolic pathway and intensity. At a low glucose concentration, the biosynthesis of 3-Met was affected by a slow growth rate and fewer cells [54]. Although previous reports have shown that a higher glucose concentration was conducive to accumulating 3-Met, an excessive concentration can change the metabolic pathway because the oxygen concentration was insufficient due to rapid cell growth [11,55]. Some studies have also shown that more ethanol and other higher alcohols may be produced at an excessive glucose concentration, which would be toxic to cells in combination with 3-Met [43].

Yeast extracts contain many vitamins and microelements, which can be used for yeast growth and affect the accumulation of yeast metabolites [8,43,56]. In the present study, 3-Met production was lower when yeast extract was absent because strain YHM-G was not able to grow well (Figure 2a). Supplementation with yeast extract increased 3-Met production compared with no addition, with the yield increasing as its concentration increased. Strain YHM-G was able to produce higher levels of 3-Met when 2–2.8 g/L of yeast extract was present with the highest production of 2.02 g/L when 2.0 g/L of yeast extract was used (Figure 2a). At a concentration of less than 2.0–2.4 g/L, yeast extract was able to be introduced as a nutrient to promote the growth of strain YHM-G [8]. While as the concentration of yeast extract increased, the bioproduction of 3-Met decreased. Previous reports have shown that almost all 3-Met is biosynthesized from L-Met as the precursor via the Ehrlich pathway, which can be affected by other nitrogen sources during this process [2,8]. Thus, under an excessive yeast extract concentration, partial yeast extraction was applied as the yeast assimilable nitrogen to cause nitrogen catabolite repression to the Ehrlich pathway, leading to a decrease in production [5,8,57,58]. An excessive addition would lead to rapid cell growth, resulting in excessive substrate consumption for cell reproduction but reducing the accumulation of metabolites. A similar observation has also been reported in previous studies where the effect of yeast extract on 3-Met bioproduction was higher at low concentrations of yeast extract [2,3,8]. It should be noted that the optimum concentration of yeast extract for 3-Met production by different yeasts varies, possibly because of their different metabolic capabilities [2,3,5].

As L-Met is the precursor for 3-Met synthesis in the Ehrlich pathway, its concentration significantly influences 3-Met synthesis by yeasts [5,59]. Like previous reports, the present study found that no 3-Met was produced when L-Met was not present, thus confirming the importance of L-Met for the biosynthesis of 3-Met and the unlikeness of successful 3-Met biosynthesis from de novo sugars by strain YHM-G (Figure 2b) [2]. More 3-Met was synthesized as L-Met increased in the fermentation medium as it is an essential precursor for 3-Met production. It has been reported that increasing the L-Met concentration can enhance the formation of 3-Met, but in the present study, the yield reached a peak of 4 g/L, and when L-Met concentrations were >4 g/L, the production of 3-Met decreased slightly (Figure 2b) [3,5,8]. The reason was that an excessive L-Met concentration would strengthen demethiolation, thereby reducing the effective L-Met concentration and intermediate metabolites in the Ehrlich pathway [3,5]. Some studies have also shown that the nitrogen content is positively correlated with the fusel acid content, so more 3-methylthiopropionic acid was produced via the Ehrlich pathway at high L-Met concentrations, which was more toxic to the cells [3,5,60,61]. Regarding the economics of production, the optimum L-Met concentration in most studies was between 1 and 5 g/L. The results of the present study were also in this range, and the differing optimal L-Met concentrations for 3-Met production with different yeasts may have been caused mainly by differences in the characteristics of the strains and also by the slight differences between the initial fermentation mediums [3,5,6,8].

Although our results have confirmed the importance of L-Met for 3-Met synthesis by strain YHM-G, the question arises as to when adding L-Met could promote 3-Met synthesis. At present, no information detailing the optimal time for L-Met addition has been reported. The optimal time at which L-Met should be added for 3-Met production by strain YHM-G was determined by adding it at different culture times. 3-Met production was significantly higher when L-Met was added after 24 h of culture compared with adding at the start of culture, with no significant increase in production at 12 or 36 h but lower production at other times (Figure 2c). The results also showed that 3-Met production decreased gradually as the time of addition was greater than 24 h (Figure 2c). This was due to the overall effect of several factors: (1) L-Met is not only required for 3-Met production but is also an amino acid required for optimum growth of yeast; (2) adding L-Met later would shorten the time available for the transformation and utilization of L-Met to 3-Met; (3) the conversion of L-Met to 3-Met via the Ehrlich pathway relies on either ATP or the electrochemical gradient so that adding L-Met at the beginning of fermentation would inhibit cell growth and adding too late would lead to insufficient ATP because of the high consumption of glucose and dissolved oxygen [5]; and (4) yeast extract contained in the fermentation medium can take over the role of L-Met as an amino acid for growth when added later. For strain YHM-G, when L-Met was added at 24 h of culture, L-Met can be fully utilized as a precursor and amino acid, so that the state of the cells and energy storage in the system was optimal.

Studies have shown that fluctuations in the environmental pH can stress microorganisms, thus changing the permeability of cell membranes and the activity of enzymes. The pH can change the state of nutrients in the system and affect the utilization of nutrients by microorganisms, resulting in different growth states and products for the microorganisms [8,25,62,63]. The optimum pH for 3-Met production was measured over a range from pH 3 to 7. The yield of 3-Met from strain YHM-G increased rapidly at a pH value between 3 to 4 and was optimal at a pH value between 4.0 and 5.5 (Figure 2d). The yield decreased with increasing pH when the pH was ≥6 (Figure 2d), which was consistent with the study of Seow et al. [2] but not with that of Matthew et al. [5]. The result showed that an initial of pH 4.0–5.5 would result in a final pH between 3.34–3.74, while when the initial pH was set at 3.0–3.5 and 6.0–7.0 and the final pH was 2.78–3.15 and 4.2–6.2, respectively. In addition, the result also illustrated that the final fermentation biomass was positively correlated with the initial pH (Figure 2d). Apparently, the optimal pH for 3-Met production was slightly lower than the optimal pH for the growth of strain YHM-G. Thus, the reason for these results was that enzyme activity in the Ehrlich pathway for 3-Met production was probably optimal under slightly acidic conditions due to its long-term survival in the slightly acidic environment of *Baijiu* making [2,5].

Temperature is an important factor affecting the growth of microorganisms and the production of metabolites. If the temperature is too low, microbial growth will be greatly compromised; an excessive temperature may lead to cell death or enzyme inactivation, discouraging the accumulation of the end-products [8,64,65]. Because of the growth temperature range of strain YHM-G, the optimum temperature for 3-Met production was measured from 20 to 40 °C. The 3-Met production increased as the temperature of incubation increased from 20 to 32 °C (Figure 2e), which was consistent with the study by Seow et al. [2]. As previously reported, the relationship between temperature and 3-Met production was not linear [2]. The increases in 3-Met formation were large, at 300% and 150%, for an increase in temperature of 4 °C, from 20 to 24 °C, and from 24 to 28 °C (Figure 2e), respectively, which was not consistent with a previous study [2], mainly because of the different strains. The highest yield of 3-Met occurred at 32 °C (Figure 2e), similar to *K. lactis* KL71 [2], which was consistent with their optimum growth temperature [2,5]. At this temperature, growth was faster and the enzymes for 3-Met biosynthesis may have exhibited a higher activity [2]. A reduction in the growth rate and activity of enzymes related to 3-Met synthesis led to a decrease in 3-Met production at high temperatures [8,66].

Although yeasts are facultative anaerobic microorganisms, their growth and most biological metabolic activities benefit from aerobic conditions [5,43]. The metabolic pathways for different metabolites differ and the oxygen concentration required also varies [25,43]. In shaking flask fermentation, the shaking speed and loading volume are the two parameters that most directly affect oxygen concentration during the fermentation process. They both regulate the production of metabolites by affecting oxygen concentration [43]. Therefore, the effect of shaking speed and loading volume on 3-Met synthesis were investigated. The results showed that 3-Met produced by strain YHM-G first increased then decreased with increasing shaking speed or loading volume (Figure 2f). With increasing shaking speed or decreasing loading volume, the oxygen concentration in the fermentation system rapidly increased growth and invigorated the metabolism, significantly increasing the yield of 3-Met [5]. When the shaking speed was 135 rpm or the loading volume was 75 mL/250 mL, the yield of 3-Met reached its highest value, at almost 10 times that achieved in an unaerated fermentation (shaking speed = 0 rpm), which was consistent with a previous study (Figure 2f) [2]. With a continuing increase in shaking speed or decrease in loading volume, the 3-Met production decreased (Figure 2f). The possible reasons for this were that the increase in shaking speed or the decrease in loading volume increased the shear force and damaged the cells, thus affecting the metabolism of the yeast, and the consequent excessive oxygen concentration caused more methional produced in the Ehrlich pathway to be oxidized to 3-methylthio-1-propanoic acid with less available for producing 3-Met [3]. Thus, it is very important to maintain an appropriate oxygen concentration to ensure the cell biomass and reaction direction in the Ehrlich pathway by regulating the shaking speed and loading volume to ensure a high yield of 3-Met.

Inoculum size is also an important parameter often considered when investigating the biosynthetic metabolites during microbial fermentation. It affects not only the yield of the target products but also the fermentation cycle [67]. An appropriate inoculum size can shorten the fermentation cycle while maintaining the yield. A low inoculum volume, such as 0.1% (the cell density was adjusted to a concentration of 1 × 10^6^ cells/mL by using a blood cell counting plate), can lead to low 3-Met production owing to the fact that longer delay period may lead to lower biomass, whereas a high inoculum volume, such as 6.4%, would also decrease the yield because excessive cell growth may lead to insufficient oxygen supply (Appendix A) [67]. There was no significant difference in 3-Met production when the inoculum size was between 0.2% and 3.2% and the yield was higher because of the superior growth rate and biomass and sufficient nutrients (Appendix A) [43]. Many previous studies have shown that the optimal inoculum size was also in this range, for example, *P. kudriavzevii* YF1702 for producing 2-phenylethanol [43] and *Bacillus licheniformis* NCIMB 8059 for producing 2,3-butanediol [68].

The cell membrane is an important component of microorganisms as it maintains cell structure and activity. It is also an important channel for material and energy transfer between cells and the surrounding environment. Changes in the permeability of the cell membrane would cause the absorption of nutrients and the secretion and extravasation of intracellular substances, thus affecting cell metabolism. The cell membrane consists of a phospholipid bilayer, so some amphiphilic surfactants, such as Tween and Triton, would affect its permeability and microbial metabolites [69,70]. Previous studies have shown that different surfactants have different effects on different metabolites of the same microorganism or on the same metabolites of different microorganisms [43]. Therefore, it was necessary to study the effect of surfactant type and concentration on 3-Met production by strain YHM-G, as no information is available at present. The results showed that both Triton X-100 and the Tween series were able to promote 3-Met production from strain YHM-G in a similar way, while glycerol and Triton X-114 inhibited 3-Met production (Figure 2g). This was because different types of surfactants had different effects on the cell membrane of strain YHM-G, which also confirmed previous reports on the effects of surfactants on microbial metabolism [43]. Tween-80 led to the maximum yield of 3-Met at 20.3% higher than the control (Figure 2g). Varying the Tween-80 concentration had little effect on 3-Met production (Appendix A), possibly because Tween-80 was compatible with the cells of strain YHM-G; even at high concentrations it still increased cell membrane permeability and promoted the dissolved oxygen level in the cells but did not permeabilize the cell membrane [70]. From considering economic factors, a lower Tween 80 concentration was chosen.

As the fermentation cycle extends, metabolites continuously accumulate, but during this process, with the consumption of nutrients, certain harmful substances also accumulate and environmental growth factors, such as pH, can change and microbial metabolic stress reactions will occur, utilizing some metabolites and converting them into other substances, resulting in a reduction in metabolites [71]. Therefore, a suitable fermentation cycle is critical for successful fermentation. The results presented in Figure 2h show the effect of culture time on 3-Met production by strain YHM-G. No 3-Met was detected during the first 12 h, possibly because the strain was multiplying during this period (Figure 2h). This may also be a reason for optimizing the time when L-Met is added or because adding L-Met at the initial stage of fermentation may impose a certain pressure on cell reproduction and growth. Previous reports have shown that the production of 3-Met may start during the initial growth phase of *K. lactis*, which was different from strain YHM-G [72]. The 3-Met concentration increased rapidly between 12 and 48 h then the production level remained essentially unchanged between 48 and 96 h (Figure 2h), which was consistent with *S. cerevisiae* EC-1118 [8]. It was observed that 3-Met was produced mainly during the early logarithmic stage of yeast growth by analyzing the growth curve and the 3-Met production curve. Production did not necessarily start if or when the growth of strain YHM-G was limited, which was consistent with a previous study [72]. 3-Met production also mainly occurred at a pH of 3.6, which may be conducive to its synthesis and was similar to results on the initial pH of the medium (Figure 2d and h).

### 3.5. Optimization of the Production Conditions by PB Design

Single factor does not consider the interaction between factors. In order to optimize fermentation conditions for YHM-G to produce 3-Met more scientifically, PB design was carried out according to the above single factor results to select the factors that have a greater impact on 3-Met production. A matrix of fifteen runs with eleven factors along with the corresponding responses for 3-Met production was designed. Changes in 3-Met production ranged from 1.00 g/L to 2.60 g/L with the test conditions (Table 6), indicating that these factors affected 3-Met production. A polynomial equation E1 that explained 3-Met production was fitted to the results:Y = 1.45 + 0.0567 X_1_ − 0.1236 X_2_ + 0.2246 X_3_ + 0.1285 X_4_ − 0.0373 X_5_ + 0.0186 X_6_ − 0.2471 X_7_ − 0.0202 X_8_ − 0.1368 X_9_ + 0.1270 X_10_ + 0.0228 X_11_

where Y is 3-Met concentration and X_1_, X_2_, X_3_, X_4_, X_5_, X_6_, X_7_, X_8_, X_9_, X_10_ and X_11_ represent shaking speed, temperature, glucose concentration, initial pH, loading volume, inoculum size, time point of L-Met addition, culture time, yeast extract concentration, L-Met concentration and Tween-80 concentration, respectively.

According to the analysis of variance (ANOVA) using Fisher’s test, the model was significant as the *p* value of the model is significant (*p* = 0.0011). The coefficient of determination (*R*^2^) was 0.9998, suggesting the model could be used to guide further study. Based on the Pareto chart (Appendix A), all eleven factors examined significantly influenced 3-Met production at the 5% level of significance. X_1_, X_3_, X_4_, X_6_, X_10_ and X_11_ had positive coefficients, whereas the other five factors showed negative coefficients (Table 2). Thus, all eleven factors merit further study. Additionally, six positive factors should be enhanced and the others should be reduced in subsequent experiments, which positively affect 3-Met production.

### 3.6. Optimization of the Production Conditions by Steepest Ascent Path Design

Based on the above regression analysis of PB design, a design matrix of six runs with eleven factors, with their step lengths as the steepest ascent method and experimental experience, according to Equation E1, were used as the steepest ascent path design (Table 3). The results showed that 3-Met production increased along the path and the yield plateaued at run five with the highest response of 2.87 g/L under cultivation for 48 h at 28 °C, 210 rpm, pH 6, 40 g/L of glucose, 0.8 g/L of yeast extract, 6 g/L of L-Met, 2.5 g/L of Tween-80 and a time point of L-Met addition of 6 h, with a loading volume of 50 mL/250 mL and 0.5% of inoculum size. Thus, the fifth set of tests was used as the central point of the RSM.

### 3.7. Optimization of the Production Conditions by RSM Design

According to the results of PB and the steepest ascent path design, a three-factor (time point of L-Met addition, glucose concentration and yeast extract concentration) and three-level BBD were used to optimize the best fermentation conditions for 3-Met production by YHM-G. Various maximum and minimum levels of three factors were used for the 15 experimental runs in Table 4. The results showed considerable variation in 3-Met production with a maximum of 3.04 g/L in test number 10 and a minimum of 1.04 g/L in test number 4.

A second order polynomial equation E2 was obtained to explain 3-Met production by applying multiple regression analysis on the experimental data:Y = 2.95 − 0.1845 A + 0.2781 B + 0.2906 C + 0.0356 AB − 0.0135 AC − 0.1315 BC − 0.4765 A^2^ − 0.4564 B^2^ − 0.5955 C^2^
where Y is the predicted response and A, B and C are the time point of L-Met addition, glucose concentration and yeast concentration, respectively. 

The ANOVA (*F*-test) shows that the model was statistically significant. The *R*^2^ was 0.9709, indicating that the sample variation of 97.09% for 3-Met production was attributed to the independent variables and only about 2.93% of the total variation was unexplained by the model (Table 7). Furthermore, the model terms of A (the time point of L-Met addition); B (glucose concentration); C (yeast extract concentration); and A^2^, B^2^ and C^2^ were significant at the 5% level, and the lack of fit was not significant with a *p* value of 0.2658 (Table 7). Thus, the model was adequate for prediction with the range of factors used.

The *p* values and coefficients suggested the independent B (glucose concentration), and C (yeast extract concentration) had a significant positive effect on 3-Met production (Table 7), indicating that they could cause an increase linearly in 3-Met production. Additionally, factor A (time point of L-Met addition) significantly negatively affected 3-Met production, indicating that factor A caused 3-Met production to decrease (Table 7). The quadric term of A, B and C also had a significant effect, whereas there was no evidence of interaction among them (Table 7).

The three-dimensional (3D) response surface plots for 3-Met production described by the above model were drawn to illustrate the effects of the independent variables and to depict the interactions between two variables by keeping the third variables at their zero levels (Figure 3). As shown in Figure 3a, 3-Met production increased first then decreased gradually as the time point of L-Met addition or glucose concentration increased, and the higher yield of 3-Met was obtained when time point of L-Met addition was at 4.2–5.6 h and glucose concentration was in the range of 41.3–43.8 g/L. The similar trend of 3-Met production was observed in Figure 3b as time point of L-Met addition or yeast concentration increased, and the yield of 3-Met was higher 3.0 g/L when time point of L-Met addition was in the range of 4.4–5.2 h and yeast concentration was in the range of 0.88–0.92 g/L. In addition, 3-Met production increased then decreased as glucose concentration or yeast concentration increased, and more than 3.0 g/L of 3-Met was obtained by YHM-G when glucose concentration was in the range of 40.7–44.6 g/L and yeast concentration was in the rage of 0.82–0.95 g/L (Figure 3c).

The maximum predicted value of 3-Met production was 3.03 g/L derived by the Design expert 11 software with the following critical values: A (time point of L-Met addition) = 4.9 h, B (glucose concentration) = 42.7 g/L and C (yeast extract concentration) = 0.9 g/L. According to the results of the statistically designed experiments, the optimized process parameters were: 42.7 g/L glucose, pH 6, 0.9 g/L yeast extract, 6 g/L L-Met, 28 °C, shaking speed 210 rpm, loading volume 50 mL/250 mL, inoculum size 0.5% (*v/v*), time point of L-Met addition 4.9 h, culturing period 48 h and 2.5 g/L Tween-80. Validation experiments were performed at the optimal level, and the yield of 3-Met was 3.16 g/L, which was 88.1% higher than the value before optimization (1.68 g/L, Table 5). Additionally, YHM-G has obvious advantages in terms of yield compared with other reported 3-Met-producing natural yeast strains (Table 1).

### 3.8. Aroma Production

Flavor substances affect food quality to a certain extent, with the unique flavor characteristics of fermented foods being mainly transformed from the ingredients in the substrate by microorganisms [73]. Thus, microorganisms with excellent flavor-producing characteristics have certain advantages for producing fermented foods. Strain YHM-G is a flavor-producing yeast screened from the *Baijiu*-making environment. The analysis of its flavor-producing characteristics would encourage its application in fermented foods. The results showed that a total of 23 flavor substances, including seven alcohols, five phenols, three esters, two alkanes, one ketone, one aldehyde, one olefin, one furan, one sulfur-containing compound and one nitrogen-containing compound, were detected in SHM after culturing strain YHM-G for 48 h (Table 8). Compared with the volatile components in SHM, there were fewer common substances in the fermentation broth and SHM, except for phenylethanol, dibutyl phthalate and benzothiazole. The content changes in two of these three flavor substances, dibutyl phthalate and benzothiazole in the fermentation broth would have been derived from SHM, but for phenylethanol, unlike SHM, most arose mainly from de novo synthesis by strain YHM-G through the metabolism of sugars [43]. Ethyl palmitate, ethyl trans oleate, ethyl linoleic acid, aniline, 2,4-dimethylbenzaldehyde and 2-methylpyrazine found in SHM was not detected in the fermentation broth, so they may have been converted into other flavor substances after fermentation by strain YHM-G, such as 2,4-dimethylbenzaldehyde being converted into 2,4-di-t-butylphenol. Similarly, flavor substances, which was not found in SHM were present in the fermentation broth. These substances include isopentanol, citronellol, p-vinyllignocerol and 2,3-dihydrobenzofuran, which were produced by strain YHM-G metabolizing substances in SHM. Notably, no 3-Met was detected in the fermentation broth since strain YHM-G could not produce it without the L-Met precursor, which was consistent with the results on optimizing the amount of L-Met optimization (Figure 3b). This observation further demonstrated that 3-Met was produced by strain YHM-G via the Ehrlich pathway not via a de novo pathway from sugars, similar to most previous reports [74]. Although strain YHM-G could not synthesize 3-Met without the L-Met precursor, it could be produced by the strain by constructing a workable synthetic microbial community, as successfully reported [7]. Among the flavor substances produced by strain YHM-G, higher alcohols, such as isobutanol, isoamyl alcohol and 2-propyl-1-pentanol, have pine, alcoholic and mushroom aromas, which can be important contributions to the flavor characteristics of fermented food [75]. Terpene alcohols, such as citronellol, trans-nerolidol and farnesol, can impart floral and fruit aromas to fermented food, and provide certain physiological functional characteristics [76,77]. Phenylethanol and its corresponding ester (phenylethyl acetate) can impart rose and honey aromas to fermented food [78]; C-nonanolactone a coconut aroma [79]; and phenolic substances produced by strain YHM-G, such as guaiacol and 4-ethyl-2-methoxyphenol, fruit, flower and sweet aromas, with antioxidant properties for food [78,80]; and 2,3-dihydrobenzofuran and N,N-dibutyl formamide can give foods a roasted flavor [81,82]. In conclusion, strain YHM-G has good potential for application in fermented foods.

## 4. Conclusions

In this study, four strains of yeast that produced a high yield of 3-Met were isolated from the *Baijiu*-producing environment. Of these, one strain of yeast, YHM-G, produced the most 3-Met. This strain was later identified as *H. burtonii*, which had not been previously reported to produce 3-Met. The strain exhibited high tolerance to sodium chloride, glucose and 3-Met. The yield of 3-Met was 3.16 g/L, which, after optimization by single factor design, PB design, steepest ascent path design and RSM design, was 1.88 times higher than that of the original culture used. The strain YHM-G can also produce many other flavor compounds, such as alcohols, phenols and esters, which are considered important for assuring the quality of many traditional foods. Thus, the *H. burtonii* strain YHM-G has potential applications for improving the quality of some traditional foods.

## Figures and Tables

**Figure 1 foods-11-03616-f001:**
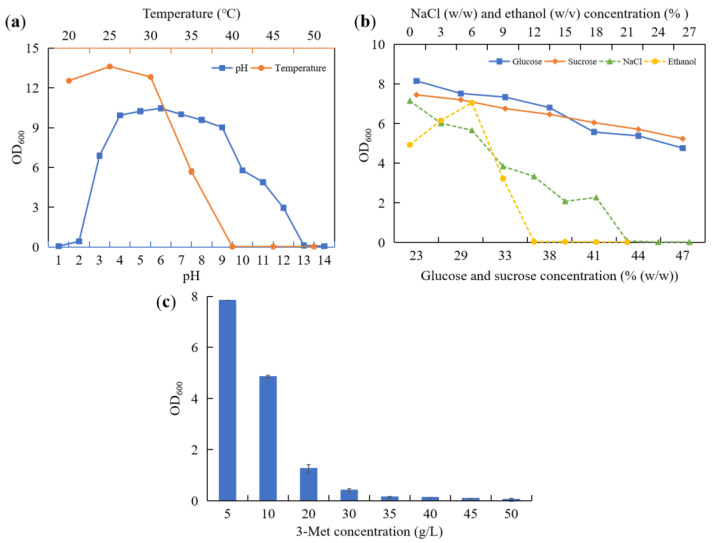
The range of growth temperature and pH (**a**) and the tolerance of glucose, sucrose, NaCl and ethanol (**b**) and 3-Met (**c**) of strain YHM-G.

**Figure 2 foods-11-03616-f002:**
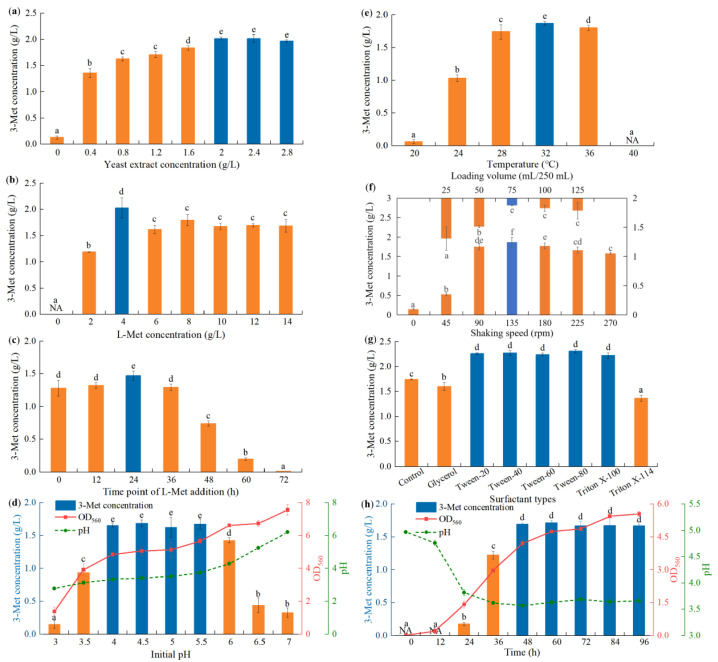
Effect of yeast extract concentration (0, 0.4, 0.8, 1.2, 1.6, 2.0 and 2.4 g/L) (**a**), L-Met concentration (0, 2, 4, 6, 8, 10, 12 and 14 g/L) (**b**), time point of L-Met addition (0, 12, 24, 36, 48, 60 and 72 h) (**c**), initial pH (3.0, 3.5, 4.0, 4.5, 5.0, 5.5, 6.0, 6.5 and 7.0) (**d**), temperature (20, 24, 28, 32, 36 and 40 °C) (**e**), shaking speed and loading volume (0, 45, 90, 135, 180, 225, 270 rpm and 25, 50, 75, 100 and 125 mL/250 mL) (**f**), surfactant types (control, glycerol, Tween-20, Tween-40, Tween-60, Tween-80, Triton X-100 and Triton X-114) (**g**) and time (0, 12, 24, 36, 48, 60, 72, 84 and 96 h) (**h**) on 3-Met concentration. Same letters in the column indicates that the data do not differ significantly at 5% probability by Tukey’s test.

**Figure 3 foods-11-03616-f003:**
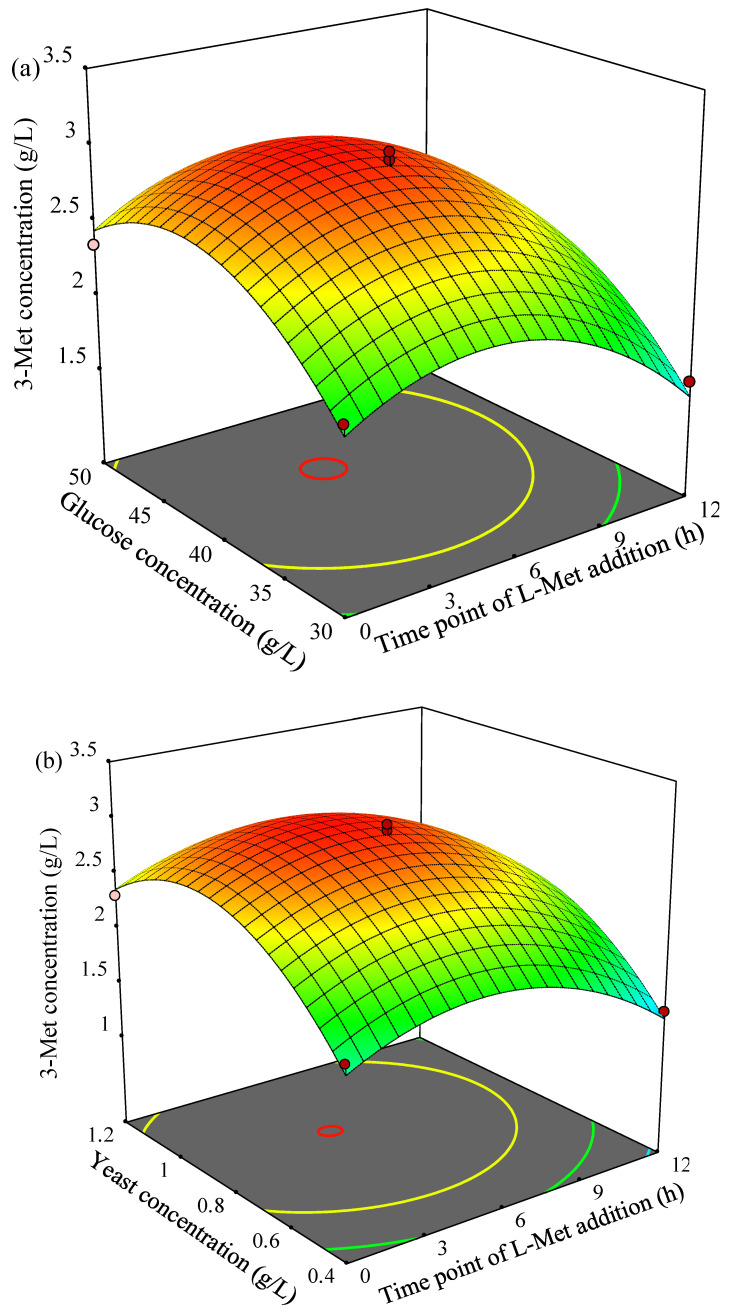
The 3D surface interaction of variable (time point of L-Met addition, glucose concentration and yeast extract concentration) on the 3-Met concentration–response using the BBD. (**a**) Time point of L-Met addition and glucose concentration; (**b**) time point of L-Met addition and yeast extract concentration; (**c**) glucose concentration and yeast extract concentration.

**Table 1 foods-11-03616-t001:** Summary of 3-Met yield in natural strains.

Classify	Strain	Origin	Culture Institution	L-Met Concentration (g/L)	3-Met Yield (mg/L)	Reference
Yeasts	*S. cerevisiae* SC408	-	China Agr Univ, Beijing 100193, China	4	1600	[11]
	*S. cerevisiae* EC-1118	-	Lallemand Inc., Ontario, Canada	1.5	190	[3]
	*S. cerevisiae* EC-1118	-	Lallemand Inc., Ontario, Canada	3	240.7 ± 17.4	[3]
	*K. lactis* KL71	-	Danisco, Singapore	4.5	990.1 ± 49.7	[2]
	*D. hansenii* DH47(8)	Cheese-ripening	UMR782 GMPA, AgroParisTech INRA, INRA Centre de Biotechnologies Agro-Industrielles, 78850, Thiverval Grignon, France	1	13.51 ± 4.57	[12]
	*K. lactis* KL640	Cheese-ripening	UMR782 GMPA, AgroParisTech INRA, INRA Centre de Biotechnologies Agro-Industrielles, 78850, Thiverval Grignon, France	1	14.51 ± 0.35	[12]
	*S. cerevisiae* SC45(3)	Cheese-ripening	UMR782 GMPA, AgroParisTech INRA, INRA Centre de Biotechnologies Agro-Industrielles, 78850, Thiverval Grignon, France	1	25.70 ± 2.02	[12]
	*Bulleromyces albus* 6655	USA, dairy atmosphere	Industrial Yeasts Collection DBVPG of the Dipartimento di Biologia Vegetale e Biotecnologie Agroambientali, Sezione di Microbiologia Applicata of the Università di Perugia	0.5	84	[13]
	*Cryptococcus magnus* 6692	Portugal, human skin	Industrial Yeasts Collection DBVPG of the Dipartimento di Biologia Vegetale e Biotecnologie Agroambientali, Sezione di Microbiologia Applicata of the Università di Perugia	0.5	101	[13]
	*C. curvatus* 6206	The Netherlands, sputum	Industrial Yeasts Collection DBVPG of the Dipartimento di Biologia Vegetale e Biotecnologie Agroambientali, Sezione di Microbiologia Applicata of the Università di Perugia	0.5	229	[13]
	*C. diffluens* 6234	Air	Industrial Yeasts Collection DBVPG of the Dipartimento di Biologia Vegetale e Biotecnologie Agroambientali, Sezione di Microbiologia Applicata of the Università di Perugia	0.5	399	[13]
	*Rhodosporidium toruloides* 6740	Sweden, wood pulp	Industrial Yeasts Collection DBVPG of the Dipartimento di Biologia Vegetale e Biotecnologie Agroambientali, Sezione di Microbiologia Applicata of the Università di Perugia	0.5	57	[13]
	*Rhodotorula acuta* 7028	Japan, grape must	Industrial Yeasts Collection DBVPG of the Dipartimento di Biologia Vegetale e Biotecnologie Agroambientali, Sezione di Microbiologia Applicata of the Università di Perugia	0.5	40	[13]
	*S. cerevisiae* JNY1	*Daqu* and fermented grain	Key Laboratory of Industrial Biotechnology of Ministry of Education, State Key Laboratory of Food Science and Technology, School of Biotechnology, Jiangnan University, Wuxi 214122, China	Sorghum extract medium	0.17 ± 0.01	[7]
Bacteria	*O. oeni IOEB* 8908	Wine	The Faculty of Oenology (Bordeaux 2 University)	1	4.04	[14]
	*O. oeni IOEB* 8413	Wine	The Faculty of Oenology (Bordeaux 2 University)	1	3.71	[14]
	*O. oeni IOEB* 8406	Wine	The Faculty of Oenology (Bordeaux 2 University)	1	3.92	[14]
	*Lactobacillus paracasei*	Cheddar cheeses	Hannah Research Institute, Ayr KA6 6HL, Scotland, UK	-	101 ± 78.9	[15]
	*L. curvatus*	Cheddar cheeses	Hannah Research Institute, Ayr KA6 6HL, Scotland, UK	-	112 ± 19.3	[15]
	*L. plantarum*	Cheddar cheeses	Hannah Research Institute, Ayr KA6 6HL, Scotland, UK	-	48.3 ± 15.4	[15]
	*L. brevis*	Cheddar cheeses	Hannah Research Institute, Ayr KA6 6HL, Scotland, UK	-	135	[15]
	*L. rhamnosus*	Cheddar cheeses	Hannah Research Institute, Ayr KA6 6HL, Scotland, UK	-	278	[15]
	*L. lactis*	Cheddar cheeses	Hannah Research Institute, Ayr KA6 6HL, Scotland, UK	-	75.6 ± 61.2	[15]
Co-culture of yeasts and bacteria	*S. cerevisiae* JNY1, *Bacillus velezensis* JNB3 and *Lentilactobacillus buchneri* JNL1	*Daqu* and fermented grain	Key Laboratory of Industrial Biotechnology of Ministry of Education, State Key Laboratory of Food Science and Technology, School of Biotechnology, Jiangnan University, Wuxi 214122, China	Sorghum extract medium	0.45 ± 0.04	[7]

Note: “-”, no details.

**Table 2 foods-11-03616-t002:** Levels of the variables and statistical analysis in PB design for 3-Met production.

Code	Variable	Low Level (−1)	High Level (+1)	Effect (EXi)	*F* Values	*p* Values	Rank	Significance
X_1_	Shaking speed (rpm)	90	180	0.11338	180.80	0.0055	7	**
X_2_	Temperature (°C)	28	36	−0.247247	859.76	0.0012	6	**
X_3_	Glucose concentration (g/L)	20	40	0.44919	2837.78	0.0004	2	***
X_4_	Initial pH	4.0	5.0	0.257057	929.34	0.0011	4	**
X_5_	Loading volume (mL)	50	100	−0.0746967	78.47	0.0125	8	*
X_6_	Inoculum size (%)	0.1	0.4	0.03717	19.43	0.0478	11	*
X_7_	Time point of L-Met addition (h)	12	36	−0.49425	3435.67	0.0003	1	***
X_8_	Culture time (h)	36	60	−0.0403067	22.85	0.0411	10	*
X_9_	Yeast extract concentration (g/L)	1.6	2.4	−0.27358	1052.66	0.0009	3	***
X_10_	L-Met concentration (g/L)	2	6	0.254017	907.49	0.0011	5	**
X_11_	Tween-80 concentration (g/L)	1	3	0.0456267	29.28	0.0325	9	*

Note: “*” Significant at 5% level (*p* < 0.05); “**” Significant at 1% level (*p* < 0.01); “***” Significant at 0. 1% level (*p* < 0.001).

**Table 3 foods-11-03616-t003:** Experimental designs and the results of steepest ascent for 3-Met production.

Groups	Shaking Speed (rpm)	Temperature (°C)	Glucose Concentration (g/L)	Initial pH	Loading Volume (mL/250 mL)	Inoculum Size (%, *v/v*)	Time Point of L-Met Addition (h)	Time (h)	Yeast Concentration (g/L)	L-Met Concentraion (g/L)	Tween-80 Concentration (g/L)	3-Met Concentration (g/L)
1	90	36	20	4	100	0.1	30	96	2.4	2	0.5	0.85
2	120	34	25	4.5	87.5	0.2	24	84	2	3	1	1.17
3	150	32	30	5	75	0.3	18	72	1.6	4	1.5	1.84
4	180	30	35	5.5	62.5	0.4	12	60	1.2	5	2	2.68
5	210	28	40	6	50	0.5	6	48	0.8	6	2.5	2.87
6	240	26	45	6.5	37.5	0.6	0	36	0.4	7	3	0.87

**Table 4 foods-11-03616-t004:** The BBD design and the responses of the dependent variables.

Test Number	Time point of L-Met Addition (h)		Glucose Concentration (g/L)		Yeast Concentration (g/L)		3-Met Concentration (g/L)
	A	Code A	B	Code B	C	Code C	Y
1	6	0	30	−1	1.2	1	2.03
2	0	−1	40	0	1.2	1	2.30
3	6	0	40	0	0.8	0	2.98
4	6	0	30	−1	0.4	−1	1.04
5	6	0	50	1	1.2	1	2.49
6	6	0	40	0	0.8	0	2.83
7	6	0	50	1	0.4	−1	2.03
8	12	1	50	1	0.8	0	2.07
9	12	1	40	0	1.2	1	1.88
10	6	0	40	0	0.8	0	3.04
11	12	1	30	−1	0.8	0	1.61
12	0	−1	30	−1	0.8	0	2.03
13	0	−1	50	1	0.8	0	2.34
14	0	−1	40	0	0.4	−1	1.84
15	12	1	40	0	0.4	−1	1.48

**Table 5 foods-11-03616-t005:** The top ten yeasts for 3-Met production in the present study.

Strain Number	3-Met Production (g/L)	Strain Number	3-Met Production (g/L)
F12504	0.82 ± 0.02 ^e^	F1504	0.77 ± 0.11 ^ef^
Y1801	1.54 ± 0.04 ^b^	F1915	0.55 ± 0.09 ^f^
F9403	1.31 ± 0.09 ^c^	Y8^#^01	0.96 ± 0.07 ^d^
F3301	1.59 ± 0.05 ^ab^	F13008	0.58 ± 0.10 ^f^
YHM-G	1.68 ± 0.02 ^a^	F1914	0.77 ± 0.06 ^e^

Note: the same letters in the table indicate that the data do not differ significantly at 5% probability by Tukey’s test; Y8^#^01 is the strain number, where Y8^#^ is the *Daqu* sample number, and 01 is the first yeast obtained from the *Daqu* sample.

**Table 6 foods-11-03616-t006:** PB design matrix for evaluating factors influencing 3-Met production.

Test Number	X_1_ (rpm)	X_2_ (°C)	X_3_ (g/L)	X_4_	X_5_ (mL)	X_6_ (%)	X_7_ (h)	X_8_ (h)	X_9_ (g/L)	X_10_ (g/L)	X_11_ (g/L)	3-Met Concentration (g/L)
1	180	28	40	4	50	0.1	36	60	2.4	2	3	1.24
2	180	28	40	5	50	0.3	12	36	1.6	6	3	2.60
3	90	36	40	4	100	0.1	12	36	2.4	6	3	1.59
4	135	32	30	4.5	75	0.2	24	48	2	4	2	1.99
5	135	32	30	4.5	75	0.2	24	48	2	4	2	1.96
6	180	36	40	4	100	0.3	12	60	1.6	2	1	1.68
7	180	28	20	4	100	0.3	36	36	2.4	6	1	1.00
8	135	32	30	4.5	75	0.2	24	48	2	4	2	1.98
9	90	28	20	4	50	0.1	12	36	1.6	2	1	1.44
10	90	28	20	5	100	0.3	12	60	2.4	2	3	1.39
11	90	36	40	5	50	0.3	36	36	2.4	2	1	1.17
12	180	36	20	5	100	0.1	36	36	1.6	2	3	1.04
13	180	36	20	5	50	0.1	12	60	2.4	6	1	1.50
14	90	36	20	4	50	0.3	36	60	1.6	6	3	1.00
15	90	28	40	5	100	0.1	36	60	1.6	6	1	1.79

**Table 7 foods-11-03616-t007:** Regression coefficients and their significances for 3-Met production from the results of the BBD.

Source	Sum of Squares	DF	Mean Square	*F* Values	*p* Values	Significance
Model	4.18	9	0.4643	18.54	0.0025	**
A—Time point of L-Met addition	0.2722	1	0.2722	10.87	0.0215	*
B—Glucose concentration	0.6188	1	0.6188	24.71	0.0042	**
C—Yeast extract concentration	0.6757	1	0.6757	26.98	0.0035	**
AB	0.0051	1	0.0051	0.2029	0.6713	
AC	0.0007	1	0.0007	0.0293	0.8708	
BC	0.0692	1	0.0692	2.76	0.1574	
A^2^	0.8383	1	0.8383	33.47	0.0022	**
B^2^	0.7690	1	0.7690	30.70	0.0026	**
C^2^	1.31	1	1.31	52.28	0.0008	***
Residual	0.1252	5	0.0250			
Lack of Fit	0.1019	3	0.0340	2.91	0.2658	Not significant
Pure Error	0.0233	2	0.0117			
Cor Total	4.30	14				
	*R*^2^ = 0.9709	*R*^2^_Adj_ = 0.9185	CV = 7.42%			

Note: “*”, Significant at 5% level (*p* <0.05); “**”, significant at 1% level (*p* < 0.01); “***” Significant at 0.11% level (*p* < 0.001).

**Table 8 foods-11-03616-t008:** The volatile compounds in SHM with or without YHM-G (µg/L).

Volatile Compounds	SHM ^a^	YHM-G ^b^	Aroma Descriptors
Isobutanol	- ^c^	18 ± 3	Pine
Isopentanol	-	335 ± 27	Alcoholic, nail polish
2-Propyl-1-pentanol	-	1 ± 0	Mushroom, cream
Citronellol	-	6 ± 0	Lemon, lime
Phenylethyl alcohol	19 ± 3	212 ± 34	Rosy, honey
Trans-nerolidol	-	2 ± 0	Rose, apple
Farnesol	-	5 ± 1	Sweet
**Σ Alcohols**	**19**	**579**	
Phenylethyl acetate	-	10 ± 1	Rosy, honey
Hexadecanoic acid ethyl ester	6 ± 0	-	Cream, herb
(E)-9-Octadecenoic acid ethyl ester	3 ± 1	-	Lipid
Ethyl linoleate	2 ± 0	-	Lipid
Dihydro-5-pentyl-2(3H)-furanone	-	4 ± 1	Coconut
Dibutyl phthalate	4 ± 1	4 ± 1	No ^d^
**Σ Esters**	**15**	**18**	
2-Octanone	-	13 ± 2	Green herbaceous
Benzaldehyde	-	27 ± 4	Almond
**Σ Ketones**	-	**40**	
Guaiacol	-	2 ± 0	Fruity, flower, sweet, green grass, sauce
4-Ethyl-2-methoxyphenol	-	10 ± 0	Warm, sweet, spicy, medicinal, clove
3-Ethylphenol	-	2 ± 0	Sweet
2-Methoxy-4-vinylphenol	-	315 ± 37	Sweet, flower, fruity, cassaba
2,4-Di-tert-butylphenol	-	6 ± 0	Sweet
**Σ Phenols**	-	**335**	
Tetradecane	-	1 ± 0	Corn, popcorn, floral,
Pentadecane	-	2 ± 0	Crayons, sweet, green grass
**Σ Alkanes**	-	**3**	
Benzothiazole	83 ± 6	11 ± 1	Smoke
4-Ethenyl-1,2-dimethoxy-benzene	-	5 ± 0	Green flower, grass, pericarp
N, N-dibutyl-formamide	-	3 ± 0	Roast
Aniline	5 ± 0	-	Sweet
Methyl-pyrazine	2 ± 0	-	Roast
2,4-Dimethyl-benzaldehyde	14 ± 2	-	Semen armeniacae amarae
2,3-Dihydro-benzofuran	-	170 ± 29	Roast, sweet
**Σ Others**	**104**	**126**	
**Sum**	**138**	**1101**	

Note: “^a^”, the volatile compounds in SHM without YHM-G; “^b^”, the volatile compounds in SHM with YHM-G; “^c^”, not detected; “^d^”, no flavor.

## Data Availability

All data and materials have been provided in this manuscript.

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
