# Peer review of "Screening of Yeasts Isolated from Baijiu Environments for Producing 3-Methylthio-1-propanol and Optimizing Production Conditions"

_foods, 2022, doi:10.3390/foods11223616_

Round 1

Reviewer 1 Report

The manuscript describes the isolation of yeast strains aiming the production of 3-Met. Among all strains, the YHM-G strain was the best producer. It was identified as H. burtonii and used for the optimization of the culture parameters for the maximal production. After optimization, the 3-Met production was increased near of 2-folds. The subject is interesting, but the manuscript needs revision to improve its quality and fluidity of ideas.

1. I suggest the revision of the English language by a professional proofreading service to improve the fluidity of the text and ideas, and to correct some minor errors through the text.

2. Line 45. It was said that some microorganisms as yeasts can biodegrade L-methionine (L-Met) to produce 3-Met. I believe that the better term is biotransform.

3. Line 56. Delete the word “identified”.

4. Table 1. This table should be reorganized. There are examples of yeasts and bacteria in this table as a mixture. Please, group all yeasts and all bacteria separately.

5. The table 4 should be presented with other orientation (landscape). The nomination of each column is tight. In addition, I believe that some table could be presented as supplementary.

6. Line 100. The “g” force should be italicized.

7. Line 170. Is the difference in the production of 3-Met by the Y1801, F9403, F3301 and YHM-G statistically significant? It is necessary discussion on the production of 3-Met comparing with other wild microorganisms, especially yeasts.

8. Figure 1. The quality should be improved.

9. Line 202-206. Move this statement to the material and methods section.

10. Line 220-222. Delete.

11. Lines 271-272. It was mentioned that the YHM-G strain was able to grow at 40 g/L of 3-Met. However, when the figure 2 is observed, it is noted that the growth in this condition is extremely reduced if compared with minor 3-Met concentrations. I believe that the best way is to consider the tolerance until 20 g/L.

12. Lines 365-369. As described, the initial pH of the culture influenced the production of 3-Met. However, there is no mention on the final pH after fermentation. This is an important parameter, since the microorganism can modify the external pH through the compounds produced during the fermentation. In addition, it was mentioned that “the optimal pH for 3-Met production was slightly lower than the optimal pH for growth of strain YHM-G, possibly because enzyme activity in the Ehrlich pathway for 3-Met production was probably optimal under slightly acidic conditions”. Which is the relation between the intracellular pH and the external pH?

13. Lines 440-444. Add specific reference that supports the mention on the effect of Twenn 80 as mentioned.

14. Conclusion (lines 625-628). Delete. These data were presented as results. In fact, they are not a conclusion.

Author Response

Responded to all of the comments for reviewer

Dear reviewer:

Thank you for the time and effort that you have put into reviewing our manuscript entitled “Screening of yeasts isolated from Baijiu environments for producing 3-methylthiopropanol and optimizing production conditions” (Manuscript ID: foods-1950396). We have carefully read the thoughtful comments from you and found that these constructive suggestions have enabled us to improve our manuscript. On the basis of the enlightening questions and helpful advices, we have now completed the revision of our manuscript and uploaded a copy of the original manuscript with all the changes highlighted by using the “Track Changes” function. The itemized responses to your comments are listed below. We hope that all these collections and revisions would be satisfactory. Thanks a lot again and the responds to your comments are as following:

The manuscript describes the isolation of yeast strains aiming the production of 3-Met. Among all strains, the YHM-G strain was the best producer. It was identified as H. burtonii and used for the optimization of the culture parameters for the maximal production. After optimization, the 3-Met production was increased near of 2-folds. The subject is interesting, but the manuscript needs revision to improve its quality and fluidity of ideas.

Response: Dear reviewer, thank you very much for your recognition of our work and giving us many constructive suggestions to improve our manuscript. And, we apologize for the deficiencies in our previous manuscript, and thank you very much for giving us the opportunity to improve the quality of our manuscript again. We have made detailed modifications and replies according to your comments. We hope that the revised manuscript can meet the requirements of our journal. If there are still deficiencies, please ask you to give us further comments and opportunities for further modifications. We will continue to make modifications and meet the requirements of the journal with the minimum times of modifications as soon as possible.

  1. I suggest the revision of the English language by a professional proofreading service to improve the fluidity of the text and ideas, and to correct some minor errors through the text.

Response: Dear reviewer, thank you very much. As your advice, we ask Philip Creed, PhD, from Liwen Bianji (Edanz) (www.liwenbianji.cn/), and prof. Dong Han for editing part of the English text of this manuscript again. We hope that our revised manuscript is fluidity of the text and ideas with your help and a professional proofreading service.

  1. Line 45. It was said that some microorganisms as yeasts can biodegrade L-methionine (L-Met) to produce 3-Met. I believe that the better term is biotransform.

Response: Dear reviewer, thank you very much. We have revised it as your professional advice: Recent studies have shown that microorganisms, especially yeasts, can biotransform L-methionine (L-Met) to produce 3-Met by fermentation, an effective method of natural production (Page 2, line 45-48).

  1. Line 56. Delete the word “identified”.

Response: Thank you very much, dear reviewer. We have deleted the word “identified” with your suggestion: However, relatively few studies have reported on the ability of microorganisms to produce natural 3-Met, which mainly included a few types of yeast such as Saccharomyces cerevisiae, Kluyveromyces lactis, Debaryomyces hansenii, Yarrowia lipolytica, Zygosaccharomyces rouxi, and Williopsis species. (Page 2, line 55-59).

  1. Table 1. This table should be reorganized. There are examples of yeasts and bacteria in this table as a mixture. Please, group all yeasts and all bacteria separately.

Response: Dear reviewer, this is a very good opinion for us to present the information of 3-Met yield. We have reorganized Table 1 according to your opinion and reviewer 2 to group all yeasts and all bacteria separately (Page 4-7, line 65-66).

  1. The table 4 should be presented with other orientation (landscape). The nomination of each column is tight. In addition, I believe that some table could be presented as supplementary.

Response: Dear reviewer, it is a perfect opinion that the table 4 is presented as landscape. This makes each column appear loose. And, as your advice, some tables are presented as supplementary after analyzing degree of importance of the data (Page 12, line 191).

  1. Line 100. The “g” force should be italicized.

Response: Thank you very much, dear reviewer. We have revised as your advice: After fermentation, the cultures were centrifuged at 4 ℃ for 10 min at 13,751×g, and the clear supernatants were filtered through a 0.22-µm filter and 3-Met production was analyzed by high performance liquid chromatography (HPLC). (Page 8, line 112-115).

  1. Line 170. Is the difference in the production of 3-Met by the Y1801, F9403, F3301 and YHM-G statistically significant? It is necessary discussion on the production of 3-Met comparing with other wild microorganisms, especially yeasts.

Response: Thank you very much, dear reviewer. At the time of initial screening, we did not carry out a significant difference analysis. With your suggestion, we carried out the statistically significant analysis and found that the 3-Met content produced by YHM-G was not significantly different from that of strain F3301, but was significantly different from that of strain Y1801 and F9403. This is our negligence. Later, we will also conduct systematic research on strain F3301. In addition, we have strengthened the comparison with other strains, especially yeast strains: In the present study, four yeasts produced 3-Met at concentrations exceeding 1.0 g/L (Table 5), which were advantage over most previously yeasts, such as S. cerevisiae EC-1118, K. lactis KL71, R. toruloides 6740 and R. acuta 7028 (Table 1). (Page 14, line 241-244).

  1. Figure 1. The quality should be improved.

Response: Dear reviewer, thank you. We have revised the quality of Figure 1 (Supplementary Fig. S1).

  1. Line 202-206. Move this statement to the material and methods section.

Response: Dear reviewer, thank you very much. We have moved them to the material and methods section. And, some changes are as follows: The ribosomal large subunit 26S rRNA gene D1/D2 do-main fragment from strain YHM-G was amplified using universal primers (NL1: 5ʹ-GCATATCAATAAGCGGAGGAAAAG-3ʹ and NL4: 5ʹ-GGTCCGTGTTTCAAGACGG-3ʹ) as described by Kurtzman and Robnett. Then, the resulting sequence was submitted to the NCBI database (https://www.ncbi.nlm.nih.gov/) with accession number ON999037 and a blast search against the Nucleotide collection (nr/nt) database using the blastn program (Page 9, line 141-148). After searching at NCBI database, the ribosomal large subunit 26S rRNA gene D1/D2 domain fragment amplificated from YHM-G showed 100% sequence similarity with a partial 26S rRNA gene sequence from Hyphopichia burtonii (CP024760.1, MH867400.1, KY107885.1, KY107883.1, KY107882.1, KY107877.1, HF952839.2, KC192660.1 and MK373312.1) (Page 15, line 281-286).

  1. Line 220-222. Delete.

Response: Dear reviewer, thank you. We have adjusted this part of data to the supplementary materials (Supplementary Table S2 and Fig. S1).

  1. Lines 271-272. It was mentioned that the YHM-G strain was able to grow at 40 g/L of 3-Met. However, when the figure 2 is observed, it is noted that the growth in this condition is extremely reduced if compared with minor 3-Met concentrations. I believe that the best way is to consider the tolerance until 20 g/L.

Response: Dear reviewer, thank you very much. We have different views on this issue, and we hope that our exchange on this issue can be recognized by you. We have carefully analyzed the articles published before. Its tolerance is judged on the basis of the highest concentration that the strain can grow (Fig. 1) [1]. This is also the basis of our judgment in sugar, NaCl, ethanol studies. Therefore, we think it is more convenient to use the highest tolerance growth concentration as the basis. If using the inhibition as the basis, it is difficult to grasp how much the degree of inhibition is the basis. For this reason, this part has not been modified.

Fig. 1 Relative growth rate of S. cerevisiae CEN.PK113-7D as a function of the 3-(methylthio)-1-propanol (methionol) methionol and 3-(methylthio)-propylacetate (3-MTPA) concentration at 30 oC in inhibition medium [1].

[1] Etschmann, M.; Kotter, P.; Hauf, J.; Bluemke, W.; Entian, K. D.; Schrader, J. Production of the aroma chemicals 3-(methylthio)-1-propanol and 3-(methylthio)-propylacetate with yeasts. Appl. Microbiol. Biot. 2008, 80, 579-587.

  1. Lines 365-369. As described, the initial pH of the culture influenced the production of 3-Met. However, there is no mention on the final pH after fermentation. This is an important parameter, since the microorganism can modify the external pH through the compounds produced during the fermentation. In addition, it was mentioned that “the optimal pH for 3-Met production was slightly lower than the optimal pH for growth of strain YHM-G, possibly because enzyme activity in the Ehrlich pathway for 3-Met production was probably optimal under slightly acidic conditions”. Which is the relation between the intracellular pH and the external pH?

Response: Dear reviewer, thank you very much. In fact, the final pH value and biomass after fermentation were measured in all our experiments. However, due to the large amount of data, we only present the final pH value and biomass for time optimization, and other conditions are not presented. We have added the final pH and biomass. As shown in Fig. 2, in different initial pH fermentation, the final pH and the biomass after fermentation is different. Although the biomass is high at pH 6.0-7.0, the highest 3-Met production is not at this initial pH. When the initial pH at 4-5.5, the pH after fermentation is between 3.34-3.74, lower than the initial pH of 6.0-7.0, and the highest 3-Met production is at this initial pH. Thus, it can be seen that the production of 3-Met from YHM-G requires not only a certain biomass, but also an appropriate pH environment, which may affect the enzyme activity in the 3-Met synthesis pathway. Although the microbial organism can regulate the pH in the cell to be relatively stable, its metabolism and the activity of related enzymes in the metabolic pathway will be regulated by the environmental pH during the whole growth process, which may be due to the self-regulation of microorganisms to adapt to the corresponding environmental pH, so as to regulate the expression level of enzymes in the related metabolic pathway, thereby affecting the product content. We have revised as follows: The result showed that when an initial of pH 4.0-5.5 would result in a final pH between 3.34-3.74, while when the initial pH was set at 3.0-3.5 and 6.0-7.0, the final pH was 2.78-3.15 and 4.2-6.2, respectively. In addition, the result also illustrated that the final fermentation biomass was positively correlated with the initial pH (Fig. 2d). Apparently, the optimal pH for 3-Met production was slightly lower than the optimal pH for growth of strain YHM-G. Thus, the reason for these results was that possibly enzyme activity in the Ehrlich pathway for 3-Met production was probably optimal under slightly acidic conditions due to its long-term survival in the slightly acidic environment of Baijiu making (Page 20, line 447-458).

  1. Lines 440-444. Add specific reference that supports the mention on the effect of Twenn 80 as mentioned.

Response: Dear reviewer, thank you very much. We have added some specific references to support the mention on the effect of Tween 80 as follows: The cell membrane consists of a phospholipid bilayer, so some amphiphilic surfactants, such as Tween and Triton, would affect its permeability and microbial metabolites [69, 70]. (Page 21, line 522-524).

  1. Conclusion (lines 625-628). Delete. These data were presented as results. In fact, they are not a conclusion.

Response: Dear reviewer, thank you very much. We have done as your advice (Page 31, line 721-724).

Reviewer 2 Report

Line 55: Oenococcus oeni is a bacterium and not a yeast.

Line 53-59: I think the sentence needs rewriting. The mentioning of “Basidiomycetous yeasts” between the ascomycetous gives an unstructured impression. I would replace “Williopsis yeasts” by “Williopsis species” if it is not possible to mention the species name.

Table 1: I would propose to first mention all yeast species and afterwards mention all bacterial species (or the other way around). Please provide information on the strain numbers. It should be mentioned to which culture institution they refer to. I would be easier to read the table if the Units were unified (eg. mg/L). It is not fully clear which strain came from which origin. Debaryomyces hansenii DH47 could originate from wine or from cheese. I would propose to repeat the origin for each strain. The same applies to the collums “concentration”, “yield”, “year” and “reference”.

Line 83-85: For which purpose were the different media used?

Line 106: The authors should check if they really sequenced and analyzed the entire large subunit ribosomal 26S rRNA gene or just the D1/D2 domain. “rDNA” should be replaced by “ribosomal large subunit 26S rRNA gene”. The authors cite “Fan et al”. If the authors used the primers NL1 and NL4 and sequences the D1/D2 domain they should cite the authors (Kurtzman and Robnett, 1998) that used this method intensively for the first time. It is necessary to state how the sequences were analyzed. Was a blast search against GenBank performed? Where the sequences of the different strains compared to the type strain sequences etc.?

Line 102: the details about the HPLC method should be given in a way that would allow to repeat the analysis under the same conditions and with equal materials.

Line 107 and 109: The authors state: “Biochemical characteristics, including temperature and pH range for growth, tolerance to glucose, NaCl, ethanol and 3-Met were determined by monitoring the optical density at 560 nm (OD560) of the YPD medium, as reported previously.”

“Biochemical characteristics” is very unspecific. Actually the authors measured the growth response. Therefore, I propose to change the sentence. The growth response was measured in a certain range of the mentioned parameters. The values of the different parameters at which the growth response was tested should be given.

Line 110 and 111: I would propose to shortly mention the type of analysis used (GC-MS, HPLC-MS etc.) and some basic information how the analysis was done. What is meant by “the characteristics”

Line 113 The authors state that the 0.1 ml of inoculum were inoculated. In table 2 is given, that the inoculum was varied in a wide range and that the optimum was between 0.2 and 3.2. Please explain the discrepancy. How did the authors make sure that the cell density of the inoculum always was in in the same range?

Line 150:  I would replace “injected” by “injection”

Line 165: At least for the one strain studied in depth (YHM-G) the taxonomic assignment should be reported in more depth. How much did the D1/D2 domain differ from that of the type strain?

Line 175-201

The phenotypic characterization should be reduced to the points that are in direct connection with the main goals of the investigation. The way the physiological tests were performed should be mentioned in the materials and methods section.

Line 202 and 203

This information should be given with the applicable literature references in the Materials and Methods section.

Line 206-214

I propose to shorten this section to the blast search against GenBan and to the pairwise sequence comparison with the type strain sequence of Hyphopichia burtonii. The fact that the sequence do not show any differences is enough to postulate that the new strain belongs to this species (Kurtzman and Robnett, 1998).

If the authors want to keep the evolutionary inference section, which I do not deem necessary,  they should mention with which algorithm the alignment was produced, how sites with gaps in the alignment were treated and on which basis the strains included in the analysis were chosen.

Line 243 – 244

The authors state “similar to that 243 found in previous reports,”. Are there previous reports on the same strain. Please explain in more detail. However, the relevant literature should be cited.

Line 251: Instead of “… the degree of inhibiting the growth of strain YHM-G also increased” I would propose “…the growth rate of strain YHM-G declined.

Line 251-252:

“When the concentration of glucose, sucrose and NaCl exceeded 90%, 90% and 21%, espectively, strain YHM-G could not grow (Fig.2b)”. This statement demands more explanation as it suggests that there was growth at 90 % glucose. It would be highly unusual that a yeast species is able to grow on such high sugar concentrations. I would propose to cite some standard literature on osmotolerance of yeasts.

Line 254

Please mention the species to which strains YF1702 and YX3307 belong.

Line 307 – 310

The sense of the sentence seems not totally clear.

Line 411 – 423

The discussion about inoculum size should include a critical statement concerning the inoculated cell numbers. If my understanding is correct the cell numbers of the inocula were not known.

Author Response

Responded to all of the comments for reviewer

Dear reviewer:

Thank you for the time and effort that you have put into reviewing our manuscript entitled “Screening of yeasts isolated from Baijiu environments for producing 3-methylthiopropanol and optimizing production conditions” (Manuscript ID: foods-1950396). We have carefully read the thoughtful comments from you and found that these constructive suggestions have enabled us to improve our manuscript. On the basis of the enlightening questions and helpful advices, we have now completed the revision of our manuscript and uploaded a copy of the original manuscript with all the changes highlighted by using the “Track Changes” function. The itemized responses to your comments are listed below. We hope that all these collections and revisions would be satisfactory. Thanks a lot again and the responds to your comments are as following:

  1. Line 55: Oenococcus oeni is a bacterium and not a yeast.

Response: Dear reviewer, thank you very much. We have corrected our mistakes: However, relatively few studies have reported on the ability of microorganisms to produce natural 3-Met, which mainly included a few types of yeast such as Saccharomyces cerevisiae, Kluyveromyces lactis, Debaryomyces hansenii, Yarrowia lipolytica, Zygosaccharomyces rouxi, and Williopsis species. (Page 2, line 55-59)

  1. Line 53-59: I think the sentence needs rewriting. The mentioning of “Basidiomycetous yeasts” between the ascomycetous gives an unstructured impression. I would replace “Williopsis yeasts” by “Williopsis species” if it is not possible to mention the species name.

Response: Dear reviewer, thanks. In order to eliminate an unstructured impression, we have deleted “Basidiomycetous yeasts” from the examples listed. In addition, in your opinion, we are in favor of changing “Williopsis yeasts” to “Williopsis species”. The sentence is as follows: However, relatively few studies have reported on the ability of microorganisms to produce natural 3-Met, which mainly included a few types of yeast such as Saccharomyces cerevisiae, Kluyveromyces lactis, Debaryomyces hansenii, Yarrowia lipolytica, Zygosaccharomyces rouxi, and Williopsis species. (Page 2, line 55-59).

  1. Table 1: I would propose to first mention all yeast species and afterwards mention all bacterial species (or the other way around). Please provide information on the strain numbers. It should be mentioned to which culture institution they refer to. I would be easier to read the table if the Units were unified (eg. mg/L). It is not fully clear which strain came from which origin. Debaryomyces hansenii DH47 could originate from wine or from cheese. I would propose to repeat the origin for each strain. The same applies to the collums “concentration”, “yield”, “year” and “reference”.

Response: Dear reviewer, thank you very much for your professional comments, we have revised Table 1 as first mention all yeast species then bacterial species. We have provided the corresponding number of the strain according to the reference. If there is no corresponding number, it is because there is no corresponding strain number information in the reference. We also supply the culture institution for the strain. And, the units were unified as your advices. Due to the confusion of information, we supplemented relevant information line by line according to your opinions (Page 4-7, line 65-66, table 1).

  1. Line 83-85: For which purpose were the different media used?

Response: Dear reviewer, thank you very much. We have supplied the purpose of different media as follows: Yeast extract peptone dextrose (YPD) medium for screening and culturing strains, Wallerstein laboratory nutrient agar (WL) medium for strain morphology identification, Sorghum hydrolysate medium (SHM) for aroma compounds production were prepared as described previously (Page 8, line 95-98).

  1. Line 106: The authors should check if they really sequenced and analyzed the entire large subunit ribosomal 26S rRNA gene or just the D1/D2 domain. “rDNA” should be replaced by “ribosomal large subunit 26S rRNA gene”. The authors cite “Fan et al”. If the authors used the primers NL1 and NL4 and sequences the D1/D2 domain they should cite the authors (Kurtzman and Robnett, 1998) that used this method intensively for the first time. It is necessary to state how the sequences were analyzed. Was a blast search against GenBank performed? Where the sequences of the different strains compared to the type strain sequences etc.?

Response: Dear reviewer, thank you very much. We have checked that we sequenced and analyzed the D1/D2 domain of ribosomal large subunit 26S rRNA gene. And, we have changed “rDNA” to “ribosomal large subunit 26S rRNA gene”. And, we have corrected the citation of reference and supplemented relevant analysis methods. As you said, we have done a blast search against the Nucleotide collection (nr/nt) database using the blastn program. And the overlapping fragment of D1/D2 domain of ribosomal large subunit 26S rRNA gene was then aligned using ClustalW. A phylogenetic tree was constructed by the neighbour-joining method (MEGA version 7 software) using Kimura 2-parameter model (Kimura 1980) and tested by bootstrap with 1000 repetitions (Kumar et al. 2008). Obtained 26S rRNA genes were subjected to BLAST against those of other yeasts available in GenBank at NCBI. The revised parts are as follows: The target yeast strain YHM-G was identified by morphological and physiological characteristics, and ribosomal large subunit 26S rRNA gene D1/D2 domain sequence, as described by Fan et al (Page 9, line 118-120). The ribosomal large subunit 26S rRNA gene D1/D2 domain fragment from strain YHM-G was amplified using universal primers (NL1: 5ʹ-GCATATCAATAAGCGGAGGAAAAG-3ʹ and NL4: 5ʹ-GGTCCGTGTTTCAAGACGG-3ʹ) as described by Kurtzman and Robnett. Then, the resulting sequence was submitted to the NCBI database (https://www.ncbi.nlm.nih.gov/) with accession number ON999037 and a blast search against the Nucleotide collection (nr/nt) database using the blastn program. All reference sequences were from type or otherwise authenticated strains. And the overlapping fragment of D1/D2 domain of ribosomal large subunit 26S rRNA gene was then aligned using ClustalW. The sequence alignment was filtered in ARB to exclude gaps, missing data, ambiguous nucleotide positions, and lower cases positions. A phylogenetic tree was constructed by the neighbour-joining method (MEGA version 7 software) using Kimura 2-parameter model and tested by bootstrap with 1000 repetitions (Page 9, line 141-153).

  1. Line 102: the details about the HPLC method should be given in a way that would allow to repeat the analysis under the same conditions and with equal materials.

Response: Dear reviewer, thank you very much. We have supplied the details about the HPLC method for 3-Met determination as follows: 3-Met concentration were determined using a 1260 series HPLC (Agilent, Santa Clara, CA, USA) equipped with a ZORBAX Eclipse Plus C18 column (4.6×250 mm, 5 µm) and a UV detector. After injection of 10 µL culture sample, the column was eluted with methanol-water (30: 70, v/v) at 30 ℃ at a flow rate of 0.7 mL/min for 15 min. The UV-detector was used at a fixed wavelength of 215 nm. The chromatogram grade 3-Met (Sigma-Aldrich Chemical Co., U.S.A.) was prepared by adding 0, 1, 2, 3, 4, and 5 mg of the standard solutions (0–5 g/L) in 1.0 mL of ultrapure water, to a serial concentration to set up the calibration curve. The standard calibration curve working solution was analyzed by HPLC. The calibration curve was obtained and used for the calculation of 3-Met in the sample. The analysis was performed in triplicate (Page 13, line 203-212).

  1. Line 107 and 109: The authors state: “Biochemical characteristics, including temperature and pH range for growth, tolerance to glucose, NaCl, ethanol and 3-Met were determined by monitoring the optical density at 560 nm (OD560) of the YPD medium, as reported previously.”

“Biochemical characteristics” is very unspecific. Actually the authors measured the growth response. Therefore, I propose to change the sentence. The growth response was measured in a certain range of the mentioned parameters. The values of the different parameters at which the growth response was tested should be given.

Response: Dear reviewer, thank you very much. We have revised as follows: the growth response was measured in a certain range of parameters, including temperature (20, 25, 30, 35, 40, 45 and 50 oC), pH (1-14), glucose concentration (23, 29, 33, 38, 41, 44 and 47%, w/w), NaCl concentration (0, 3, 6, 8, 11, 13, 15, 17, 19 and 21%, w/v), ethanol concentration (0, 3, 6, 9 12, 15, 18 and 21%, v/v) and 3-Met concentration (5, 10, 20, 30, 35, 40, 45 and 50 g/L) by monitoring the optical density at 560 nm (OD560) of the YPD medium, as reported previously. And, we added the values of the different parameters (Page 9, line 154-162).

  1. Line 110 and 111: I would propose to shortly mention the type of analysis used (GC-MS, HPLC-MS etc.) and some basic information how the analysis was done. What is meant by “the characteristics”

Response: Dear reviewer, thank you very much. We have supplied the type of analysis used and some basic information for analysis as follows: The aroma compounds were analyzed by headspace solid–phase microextraction gas chromatography–mass spectrometry (HS–SPME–GC–MS) with a TraceGC TSQ 8000 Evo instrument (Thermo Fisher Scientific, Waltham, MA, United States) according to our previous report. Briefly, the aroma compounds in samples were extracted by 50/30-µm divinylbenzene/carboxen on polydimethylsiloxane (DVB/CAR on PDMS)-coated fibers. After centrifuge, 5 mL supernatant, 2 g NaCl, and 10 µL of 0.5 g/L tetraoctanol solution was put into a 15-mL headspace vial. After kept in a water bath at 50 ℃ for 10 min, a SPME needle was inserted to the vial to extract aroma compounds for 30 min at 50 ℃. Then, the extraction head was inserted into the GC-MS inlet for desorption at 250 ℃ for 5 min. The GC-MS conditions were as follows: 40 ℃ for 3 min, increased up to 100 ℃ for 5 min, then 150 ℃ for 3 min, and to 280 ℃ for 6 min. The impact energy was 70 eV, and the mass spectral scanning range was m/z 33-550. The splitless mode was conducted for these analyses. The aroma compounds that the mass spectra with positive and negative matches higher than 800 with the National Institute of Standards and Technology library are reported. The amount of each aroma compound was calculated as the ratio of the mass concentration of tetraoctanol to the mass concentration of the compound (Page 13-14, line 212-228). We are so sorry the using “the characteristics”. As your prompt, we have changed as follows: The aroma compounds produced by YHM-G in SHM were carried out as described previously (Page 9, line 163-164).

  1. Line 113 The authors state that the 0.1 ml of inoculum were inoculated. In table 2 is given, that the inoculum was varied in a wide range and that the optimum was between 0.2 and 3.2. Please explain the discrepancy. How did the authors make sure that the cell density of the inoculum always was in in the same range?

Response: Dear reviewer, thank you very much. As we mentioned in method, the cell density was adjusted to a concentration of 1×106 cells/mL by using a blood cell counting plate as to ensure the amount of inoculated cell. And, the culture conditions for seed broth were consistent. So, these measures make sure that the cell density of the inoculum was in the same range. Inspired by Reviewer 1, we analyzed the final pH and biomass after fermentation, and found that although the initial inoculation amount was different, the final pH after fermentation was similar, and the biomass showed an increasing trend. Too low an inoculum size, such as 0.1%, meant that 3-Met production was lower because the longer delay period and lower biomass led to a lower yield but at too high an inoculum size, such as 6.4%, the yield was also lower because of excessive cell growth consuming the nutrients leading to insufficient oxygen. When the inoculum size was between 0.2% and 3.2% and the yield was higher because of the superior growth rate and biomass, and sufficient nutrients (Page 21, line 503-511).

  1. Line 150: I would replace “injected” by “injection”

Response: Dear reviewer, thank you very much. We have revised as your said: After injection of 10 µL culture sample, the column was eluted with methanol-water (30: 70, v/v) at 30 ℃ at a flow rate of 0.7 mL/min for 15 min (Page 13, line 205-207).

  1. Line 165: At least for the one strain studied in depth (YHM-G) the taxonomic assignment should be reported in more depth. How much did the D1/D2 domain differ from that of the type strain?

Response: Dear reviewer, thank you very much. We have a detail for taxonomic assignment for strain YHM-G. The ribosomal large subunit 26S rRNA gene D1/D2 domain of YHM-G have a sequence similarity of 100% with a partial 26S rRNA sequence from Hyphopichia burtonii (CP024760.1, MH867400.1, KY107885.1, KY107883.1, KY107882.1, KY107877.1, HF952839.2, KC192660.1 and MK373312.1). And, according to branch clustering, strain YHM-G was closest to H. burtonii CBS:2352 (KY107882.1) and H. burtonii CBS:232.37 (MH867400.1) in the phylogenetic tree constructed using MEGA 7.0.14 software (https://www.megasoftware.net) based on the neighbor-joining statistical method (Page 15, line 281-289).

  1. Line 175-201

The phenotypic characterization should be reduced to the points that are in direct connection with the main goals of the investigation. The way the physiological tests were performed should be mentioned in the materials and methods section.

Response: Dear reviewer, thank you very much. In order to make better use of strains for subsequent production and application, it is necessary to conduct more research on strains. Among them, the analysis of physiological and biochemical characteristics is an important link to recognize the strain. Therefore, we refer to the physiological and biochemical analysis related to the strain for research. In your opinion, we have supplemented the methods related to physiology and biochemistry as follows: YHM-G was examined for colony morphology on WL plate by naked eye and cell morphology using a microscope (Nikon, Japan), respectively. Carbon and nitrogen source assimilation and sugar fermentation test were conducted in stationary cultures as sole source of carbon or nitrogen. Slanted four millilitres of urea medium which placed in slants applied for the incubated yeast colony at room temperature. Red-pink colour in the medium was considered as a positive test for urea test. For indole test, appearance of bright red and yellow color which composed after added 0.5 ml of Kovac's reagent to incubated culture at 30  ℃ for 24  h on indole test media indicated a positive and negative results respectively. After adding methyl red indicator solution to inoculated culturing media and incubation at 30  ℃ for up to 7 days, changing color to red indicate methyl red test positive- appearance. For Voges-Proskauer test after incubation of 48 h Voges-Proskauer Reagent A (5% naphthol) and Voges-Proskauer Reagent B (40% KOH) were added and change in color was observed. YHM-G was inoculated on triple suger iron agar slant and cultured at 30  ℃ for 48 h, and results were determined on the basis of color change. Citrate test was performed via inculcate Simmons Citrate Agar plates surface with yeast cultures then, incubated at 30  ℃ up to 48 h changing media colour from green to bright blue indicate positive reaction. Gelatin liquidized test was applied as that YHM-G was inoculated into tubes containing gelatin, gelatin liquefaction is the positive results. To confirm the starch hydrolysis activity, YHM-G were grown on nutrient agar containing 1% soluble starch and incubated at 30 ℃ for 72 h. Hydrolysis was observed as formation of a colorless zone around colony after flooding the plate with Gram’s iodine (Page 9, line 120-141).

  1. Line 202 and 203

This information should be given with the applicable literature references in the Materials and Methods section.

Response: Dear reviewer, thank you very much. We have supplied the references in the Materials and Methods section as your advices: The ribosomal large subunit 26S rRNA gene D1/D2 domain fragment from strain YHM-G was amplified using universal primers (NL1: 5ʹ-GCATATCAATAAGCGGAGGAAAAG-3ʹ and NL4: 5ʹ-GGTCCGTGTTTCAAGACGG-3ʹ) as described by Kurtzman and Robnett [21]. Then, the resulting sequence was submitted to the NCBI database (https://www.ncbi.nlm.nih.gov/) with accession number ON999037 and a blast search against the Nucleotide collection (nr/nt) database using the blastn program. All reference sequences were from type or otherwise authenticated strains. And the overlapping fragment of D1/D2 domain of ribosomal large subunit 26S rRNA gene was then aligned using ClustalW. The sequence alignment was filtered in ARB to exclude gaps, missing data, ambiguous nucleotide positions, and lower cases positions. A phylogenetic tree was constructed by the neighbour-joining method (MEGA version 7 software) using Kimura 2-parameter model [22] and tested by bootstrap with 1000 repetitions [23] (Page 9, line 141-153).

[21] Kurtzman, C. P.; Robnett, C. J. Identification and phylogeny of ascomycetous yeasts from analysis of nuclear large subunit (26S) ribosomal DNA partial sequences. Anton. Leeuw. Int. J. G. 1998, 73, 331-371.

[22] Kimura, M. A simple method for estimating evolutionary rates of base substitutions through comparative studies of nucleotide sequences. J. Mol. Evol. 1980, 16, 111-120.

[23] Kumar, S.; Nei, M.; Dudley, J.; Tamura, K. Mega: a biologist-centric software for evolutionary analysis of DNA and protein sequences. Brief. Bioinform. 2008, 9, 299-306.

  1. Line 206-214

I propose to shorten this section to the blast search against GenBan and to the pairwise sequence comparison with the type strain sequence of Hyphopichia burtonii. The fact that the sequence do not show any differences is enough to postulate that the new strain belongs to this species (Kurtzman and Robnett, 1998).

If the authors want to keep the evolutionary inference section, which I do not deem necessary, they should mention with which algorithm the alignment was produced, how sites with gaps in the alignment were treated and on which basis the strains included in the analysis were chosen.

Response: Dear reviewer, thank you very much. We have done as you said. We have supplied the algorithm the alignment and how sites with gaps in the alignment were treated and on which basis the strains included as follows: Then, the resulting sequence was submitted to the NCBI database (https://www.ncbi.nlm.nih.gov/) with accession number ON999037 and a blast search against the Nucleotide collection (nr/nt) database using the blastn program. All reference sequences were from type or otherwise authenticated strains. And the overlapping fragment of D1/D2 domain of ribosomal large subunit 26S rRNA gene was then aligned using ClustalW. The sequence alignment was filtered in ARB to exclude gaps, missing data, ambiguous nucleotide positions, and lower cases positions. A phylogenetic tree was constructed by the neighbour-joining method (MEGA version 7 software) using Kimura 2-parameter model [22] and tested by bootstrap with 1000 repetitions [23] (Page 9, line 148-153).

  1. Line 243 – 244

The authors state “similar to that 243 found in previous reports,”. Are there previous reports on the same strain. Please explain in more detail. However, the relevant literature should be cited.

Response: Dear reviewer, thank you very much. This is our mistake. We mean that the optimum growth temperature of YHM-G is similar to that of many other strains reported. So, we have revised as follows: The optimum growth temperature of strain YHM-G was 25 ℃, similar to most yeasts that found in previous reports, with a maximum growth temperature of 40 ℃ (Fig. 1a), which was higher than most temperatures (20-40 ℃) during the Baijiu production process (Page 17, line 320-323).

  1. Line 251: Instead of “… the degree of inhibiting the growth of strain YHM-G also increased” I would propose “…the growth rate of strain YHM-G declined.

Response: Thank you very much. We have revised as your said: As the glucose concentration, sucrose concentration and NaCl concentration increased the growth rate of strain YHM-G declined (Page 17, line 327-329).

  1. Line 251-252:

“When the concentration of glucose, sucrose and NaCl exceeded 90%, 90% and 21%, espectively, strain YHM-G could not grow (Fig.2b)”. This statement demands more explanation as it suggests that there was growth at 90 % glucose. It would be highly unusual that a yeast species is able to grow on such high sugar concentrations. I would propose to cite some standard literature on osmotolerance of yeasts.

Response: Thank you very much. This is an error in our data. When preparing sugar solution, we weighed 900 g of solid powder and added it to 1 L of water. In fact, we cannot simply use the sugar mass ratio to the solution volume as the sugar concentration. Because the sugar quality is very high, which affects the final volume. We need use the method of sugar mass ratio to the total mass, that is, the mass percentage. Thank you very much for pointing out our serious mistake. We have revised as follows: The growth response was measured in a certain range of parameters, including temperature (20, 25, 30, 35, 40, 45 and 50 oC), pH (1-14), glucose concentration (23, 29, 33, 38, 41, 44 and 47%, w/w), NaCl concentration (0, 3, 6, 8, 11, 13, 15, 17, 19 and 21%, w/v), ethanol concentration (0, 3, 6, 9 12, 15, 18 and 21%, v/v) and 3-Met concentration (5, 10, 20, 30, 35, 40, 45 and 50 g/L) by monitoring the optical density at 560 nm (OD560) of the YPD medium, as reported previously (Page 9, line 154-159). When the concentration of glucose, sucrose and NaCl exceeded 47%, 47% and 17%, respectively, strain YHM-G could not grow (Fig. 1b) (Page 17, line 330-331). And, we have supplied some standard literature on osmotolerance of yeasts as follows: In general, yeast strains have a high tolerance to sugar with strains such as Pichia kudriavzevii YF1702 and Clavispora lusitaniae YX3307 [25, 43-46] (Page 17, line 331-333).

[25] Fan, G. S.; Liu, P. X.; Chang, X.; Yin, H.; Cheng, L. J.; Teng, C.; Gong, Y.; Li, X. T. Isolation and identification of a high-yield ethyl caproate-producing yeast from Daqu and optimization of its fermentation. Front Microbiol. 2021, 12.

[43] Fan, G. S.; Cheng, L. J.; Fu, Z. L.; Sun, B. G.; Teng, C.; Jiang, X. Y.; Li, X. T. Screening of yeasts isolated from Baijiu environments for 2-phenylethanol production and optimization of production conditions. 3 Biotech. 2020, 10, 275.

[44] Praphailong, W.; Fleet, G. H. The effect of pH, sodium chloride, sucrose, sorbate and benzoate on the growth of food spoilage yeasts. Food Microbiol. 1997, 14, 459-468.

[45] Tokuoka, K.; Ishitani, T. Minimum water activities for the growth of yeasts isolated from high-sugar foods. J. Gen. Appl. Microbiol. 1991, 37, 111-119.

[46] Tokuoka, K. Sugar- and salt-tolerant yeasts. J. Appl. Bacteriol. 1993, 74, 101-110.

  1. Line 254

Please mention the species to which strains YF1702 and YX3307 belong.

Response: Thank you very much. We have supplied the information for YF1702 and YX3307 as follows: In general, yeast strains have a high tolerance to sugar with strains such as Pichia kudriavzevii YF1702 and Clavispora lusitaniae YX3307 [25, 43-46] (Page 17, line 331-333).

  1. Line 307 – 310

The sense of the sentence seems not totally clear.

Response: Thank you very much. We made a mistake in writing this sentence. We have revised them as follows: Similar observation has also been reported in previous studies where the effect of yeast extract on 3-Met bioproduction was higher at low concentrations of yeast extract (Page 19, line 392-395).

  1. Line 411 – 423

The discussion about inoculum size should include a critical statement concerning the inoculated cell numbers. If my understanding is correct the cell numbers of the inocula were not known.

Response: Thank you very much. In fact, as we mentioned in method, the cell density was adjusted to a concentration of 1×106 cells/mL by using a blood cell counting plate as to ensure the amount of inoculated cell. We have revised as your advice as follows: A low inoculum volume, such as 0.1% (the cell density was adjusted to a concentration of 1×106 cells/mL by using a blood cell counting plate), can led to low 3-Met production owing to the fact that longer delay period may lead to lower biomass, whereas high inoculum volume, such as 6.4%, would also decrease the yield because excessive cell growth may lead to insufficient oxygen supply (Page 21, line 503-511).

Thank you again for your professional advices. And, we hope that all these collections and revisions would be satisfactory. If there are still different opinions on our reply, please tell us in time, and we will reply again according to your opinions.

We look forward to your reply.

Best wishes.

Guangsen Fan

Reviewer 3 Report

This is one outstanding and relevant paper. I enjoyed reading it, but I have a few suggestions for the authors.

0. Think about graphical abstract; This is always a way to attract future readers.

1. Please, expand the Introduction part with state-of-the-art of the main topic/goal of the paper; 

2. Avoid using "However" in consecutive sentences;

3. Add the primary goal of the paper at the end of the Introduction through the main advantages of the further results

4. Can you provide a picture of Daqu samples? I think it is a step forward in understanding the concept of the paper.

Author Response

Responded to all of the comments for reviewer

Dear reviewer:

Thank you for the time and effort that you have put into reviewing our manuscript entitled “Screening of yeasts isolated from Baijiu environments for producing 3-methylthiopropanol and optimizing production conditions” (Manuscript ID: foods-1950396). We have carefully read the thoughtful comments from you and found that these constructive suggestions have enabled us to improve our manuscript. On the basis of the enlightening questions and helpful advices, we have now completed the revision of our manuscript and uploaded a copy of the original manuscript with all the changes highlighted by using the “Track Changes” function. The itemized responses to your comments are listed below. We hope that all these collections and revisions would be satisfactory. Thanks a lot again and the responds to your comments are as following:

This is one outstanding and relevant paper. I enjoyed reading it, but I have a few suggestions for the authors.

Response: Dear reviewer, thank you very much for your recognition of our work and your very professional suggestions. We have carefully studied your comments and made detailed modifications according to your comments. We hope that our modifications can meet your comments on our manuscript.

  1. Think about graphical abstract; This is always a way to attract future readers.

Response: Dear reviewer, thank you very much. We have revised the graphical abstract.

Graphical abstract

  1. Please, expand the Introduction part with state-of-the-art of the main topic/goal of the paper;

Response: Dear reviewer, thank you very much. We have expanded the main topic of the paper in Introduction part as follows: After determining the key enzymes that affect the synthesis of 3-Met in yeast strain, the yeast was molecular modified through gene technology (Page 8, line 70-72).

  1. Avoid using "However" in consecutive sentences;

Response: Dear reviewer, thank you very much. We have revised the whole manuscript as your advice.

  1. Add the primary goal of the paper at the end of the Introduction through the main advantages of the further results

Response: Dear reviewer, thank you very much. We have added the primary goal at the end of the Introduction part as follows: Therefore, the objective of this study was to obtain a strain with a high 3-Met yield from the Baijiu-making environment and to optimize its fermentation conditions. In this study, a yeast identified as Hyphopichia burtonii with a high yield of 3-Met, was obtained from the Baijiu-brewing environment. Several statistical methods will be combined to optimize the fermentation conditions for this strain to further enhance its 3-Met production. To the best of the author's knowledge, the strain discovered in this study is among the highest 3-Met yielding natural strains that has been previously reported (Page 8, line 79-88).

  1. Can you provide a picture of Daqu samples? I think it is a step forward in understanding the concept of the paper.

Response: Dear reviewer, thank you. We have provided a picture of Daqu samples in Graphical Abstract.

Thank you again for your professional advices. And, we hope that all these collections and revisions would be satisfactory. If there are still different opinions on our reply, please tell us in time, and we will reply again according to your opinions.

We look forward to your reply.

Best wishes.

Round 2

Reviewer 1 Report

The revised version of the manuscript was improved according to the most of suggestions. However, some points need consideration yet.

1. Some minor errors of English language can be found through the text. Please, verify.

2. Line 59. Delete the word “And”. Start the sentence as “The ability…”.

3. Table 1. There is a mixture among the yeast and bacteria strains yet. Please, verify. In addition, the column “year” must be deleted because the year can be verified during the observation of each reference mentioned.

4. Line 72. What the “gene technology” means? This mention is superficial.

5. Results and Discussion. Previously, it was asked on the statistical significance on the production of 3-Met by the Y1801, F9403, F3301 and YHM-G strains. The author’s response indicated that “the 3-Met content produced by YHM-G was not significantly different from that of strain F3301, but was significantly different from that of strain Y1801 and F9403.” However, this was not mentioned in the text. Please, add.

Author Response

Dear reviewer:

Thank you again for the time and effort that you have put into reviewing our manuscript entitled “Screening of yeasts isolated from Baijiu environments for producing 3-methylthiopropanol and optimizing production conditions” (Manuscript ID: foods-1950396). These constructive suggestions have enabled us to improve our manuscript. We have now completed the revision of our manuscript and uploaded a copy of the original manuscript with all the changes highlighted by using the “Track Changes” function. The itemized responses to your comments are listed below. Thanks a lot again.

The revised version of the manuscript was improved according to the most of suggestions. However, some points need consideration yet.

Response: Dear reviewer, thank you very much for your important comments on our manuscript. With your help, the quality of our manuscript has been improved. Thank you very much for your recognition of our work. We apologize for the shortcomings that still exist, and we will further revise our manuscript according to your comments, hoping to meet high quality requirements of this journal.

  1. Some minor errors of English language can be found through the text. Please, verify.

Response: Dear reviewer, we are very sorry that, in your opinion, we have made another revision to the language of our manuscript, and all the revisions have been marked in the manuscript.

  1. Line 59. Delete the word “And”. Start the sentence as “The ability…”.

Response: Dear reviewer, thank you very much. We have revised it as your professional advice: The ability of these natural strains to produce 3-Met is limited with yields generally less than 0.5 g/L (Table 1), therefore their application as a microbial factory in industrial fermentation has been restricted (Page 2, line 56-58).

  1. Table 1. There is a mixture among the yeast and bacteria strains yet. Please, verify. In addition, the column “year” must be deleted because the year can be verified during the observation of each reference mentioned.

Response: Thank you very much, dear reviewer. Yes, it is. The information in the reference was a mixture of yeasts and bacteria to coculture to produce 3-Met. And, we have deleted the column “year” as your suggestion (Page 3-5, Table 1).

  1. Line 72. What the “gene technology” means? This mention is superficial.

Response: Dear reviewer, this is a very good opinion for us to present the detail information. We have reviewed according to your opinionas follows: After determining the key enzymes that affect the synthesis of 3-Met in yeast strain, the yeast genome was molecular modified through gene technology overexpression or knockout (Page 6, line 64-66).

  1. Results and Discussion. Previously, it was asked on the statistical significance on the production of 3-Met by the Y1801, F9403, F3301 and YHM-G strains. The author’s response indicated that “the 3-Met content produced by YHM-G was not significantly different from that of strain F3301, but was significantly different from that of strain Y1801 and F9403.” However, this was not mentioned in the text. Please, add.

Response: Dear reviewer, thank you very much, we have added in the text as your advice and added the significant difference analysis in Table 1. We have revised them as follows: Among the high 3-Met yielding strains evaluated in this study, the 3-Met yield of YHM-G was not significantly higher than strain F3301, but was significantly higher than strain Y1801 and F9403, and the yeast strain YHM-G produced 1.68 g/L of 3-Met among the highest 3-Met yield in natural microorganisms (Table 1) (Page 11, line 232-236).

Thank you again for your professional advices. And, we hope that all these collections and revisions would be satisfactory. If there are still different opinions on our reply, please tell us in time, and we will reply again according to your opinions.

We look forward to your reply.

Best wishes.

Guangsen Fan
